# A systematic mapping review of therapeutic clinical trials in dengue

Tran Bang Huyen[1,2*], Angela McBride[3*], Tun-Linn Thein[4], Khoi Minh Le[1], Tran Luu[1], Nguyen Quang Huy[1], Eli Harriss[5], Matthew J.W. Kain[6,7], Jonathan Cattrall[3], Caitlin Naylor[2,8], Ho Quang Chanh[1,2], Nguyen Lam Vuong[1,9], Daniel Munblit[10,11], Po-Ying Chia[4,12], Phung Khanh Lam[13], James A. Watson[2,8,&], Sophie Yacoub[1,3&]

1 Oxford University Clinical Research Unit, Ho Chi Minh City, Vietnam, 2 Nuffield Department of Medicine, Centre of Tropical Medicine and Global Health, University of Oxford, Oxford, United Kingdom, 3 Pandemic Sciences Institute, University of Oxford, Oxford, United Kingdom, 4 National Centre for Infectious Diseases, Singapore, Singapore, 5 Bodleian Healthcare Libraries, University of Oxford, Oxford, United Kingdom, 6 Institute of Naval Medicine, Gosport, United Kingdom, 7 Department of Clinical Sciences, Liverpool School of Tropical Medicine, Liverpool, United Kingdom, 8 Infectious Diseases Data Observatory, Oxford, United Kingdom, 9 University of Medicine and Pharmacy at Ho Chi Minh City, Ho Chi Minh City, Vietnam, 10 Kings College London, London, United Kingdom, 11 Sechenov First Moscow State Medical University, Moscow, Russia, 12 Lee Kong Chian School of Medicine, Nanyang Technological University Singapore, Singapore, Singapore, 13 National University of Singapore, Singapore, Singapore

& These authors jointly supervised this work and co-senior authors
* huyentb@oucru.org (TBH); angela.mcbride@ndm.ox.ac.uk (AC)

## Abstract

### Background

Dengue is a growing public health threat with increasing case numbers globally. Despite the substantial burden, there are no licensed therapeutics for patients with dengue. To inform the design of large-scale practice-changing clinical trials and to assess the feasibility of an individual patient data platform for meta-analysis, we conducted a systematic mapping review of clinical trials evaluating dengue therapeutics. Our aims were to characterise published and registered dengue therapeutic trials, describe their endpoints, and assess study design quality and internal validity to inform feasibility of meta-analysis and future research.

### Methods

We systematically searched Ovid MEDLINE, Ovid EMBASE, WHO ICTRP and ClinicalTrials.gov for prospective clinical trials evaluating therapeutics in patients with symptomatic dengue. Two independent reviewers screened records using Covidence. Data were extracted into a REDCap database, and risk of bias was assessed using the ROB-2 and ROBINS-I tools to describe trial design rigour. Descriptive analyses summarised the interventions, trial characteristics, study populations, and primary endpoints. This systematic review was pre-registered with PROSPERO (CRD42023469022).

**Data availability statement:** The extracted data have been included in supplementary information (S2 Data).

**Funding:** JC received a University of Oxford - Medical Research Council Doctoral Training Partnership [MR/W006731/1]. DM reports received grants from the Bill and Melinda Gates Foundation [INV-063472]. PKL received funding from the National Research Foundation, Singapore [SG Academies South-East Asia Fellowship (SASEAF) Programme]. TBH received funding from the Wellcome Trust core grant at OUCRU Vietnam [106680]. SY received funding from the Wellcome Trust [223004]. The funders had no role in study design, data collection and analysis, decision to publish, or preparation of the manuscript.

**Competing interests:** I have read the journal's policy and the authors of the manuscript have the following competing interests: DM acknowledges a leadership role in the DEN-CORE project that developed core outcome measurement set for dengue clinical trials. SY serves on the scientific committee for the Takeda dengue vaccine program and has participated in an advisory board for Novartis, for which she has received consulting fees. She is also Chair of the DSMB for a Phase 2 antiviral trial conducted by Novartis.

## Results & discussion

A total of 121 clinical studies were identified, comprising 72 published trials and 49 registered but unpublished studies. Interventions were categorised according to the authors' proposed mechanism of action: antiviral (n = 10), host-directed (HDT, n = 34), supportive (n = 31), or undefined (n = 46). Aside from the studies of supportive therapies (n = 31) and unpublished studies (n = 37) which were only reviewed for their primary outcomes, 53 publications remained for review of therapeutic efficacy. Methodological concerns were common – 24 of 53 published trials (45%) were classified as having high or critical risk of bias. Corticosteroids were the most frequently evaluated intervention, involving a total of 944 randomised patients. The primary endpoints used in both antiviral and HDT trials were highly heterogeneous, limiting comparability. The combination of methodological concerns and non-standardised endpoints precluded meta-analysis for any intervention. No single treatment had sufficient or consistent evidence to support recommendations for use in clinical practice.

## Conclusions

Our findings highlight a remarkably sparse evidence base for dengue therapeutics and a lack of standardised, clinically meaningful endpoints. These factors have hindered progress in evaluating candidate treatments and limited the potential for individual patient data meta-analyses. Large, high-quality trials - powered for harmonised and clinically relevant endpoints - are urgently needed to advance the development of effective therapies for dengue.

---

### Author summary

Dengue is a viral infection spread by mosquitoes that causes approximately 100 million symptomatic cases worldwide each year. Despite this growing global burden, there are still no approved drugs to treat the virus itself or the inflammation it causes in the body. To assess the current state of treatment options and guide future research, we systematically reviewed all clinical trials that have tested potential treatments for dengue. We also looked at whether data from past studies could be combined to reveal new insights. We identified 121 studies covering diverse treatments. The overall quality of the studies were low. Most were small, early-phase trials using inconsistent methods, which made it impossible to combine results in a meaningful way or draw firm conclusions about effectiveness. No treatment had enough consistent evidence to recommend for clinical use. Our findings show that research on dengue therapeutics has been fragmented, underpowered, and limited by non-standardised outcomes. Moving forward, the dengue research community needs to coordinate large, standardized trials. By harmonizing how we measure success, researchers can generate the robust evidence needed to improve patient care.

## Background

Dengue is an arboviral infection of major public health importance, with an estimated 96 million symptomatic infections globally each year [1]. The rise in cases over the past three decades has been driven by urbanisation, human mobility and climate change [2,3]. Climate change is lengthening transmission seasons in endemic regions, driving expansion of the mosquito vector to altitudes and latitudes where sustained transmission was not previously reported, and rendering new populations susceptible to infection [2,4]. In endemic countries within Asia and Latin America, dengue is one of the leading causes of hospital admission and outbreaks regularly overwhelm health systems. While most clinical cases of dengue resolve within 10 days, 25–40% of patients require hospital admission for monitoring, and 5–10% of hospitalised patients develop severe manifestations including shock, bleeding and organ impairment [5–7]. Despite the considerable public health burden, there are currently no licensed therapies, and treatment is limited to supportive care only.

The therapeutic goal for early dengue is to prevent the development of more severe manifestations necessitating hospitalisation, while treatment of severe dengue aims to prevent worsening of organ failure and death. Studies on the pathogenesis of dengue have highlighted two major potential therapeutic strategies: antiviral agents and host-directed therapeutics (HDTs). Viraemia peaks early in the illness (day 2–3) whereas severe manifestations occur later (days 4–6). Higher viral densities in blood and slower clearance of virus are both associated with more severe outcomes [8,9], raising the hypothesis that an effective antiviral administered during the early phase of illness may reduce progression to severe disease. HDTs may have a role in reducing progression to severe disease in those at highest risk, or improving outcomes in those with established severe disease. Accumulating evidence suggests that immunopathogenesis is a major driver of severe disease; higher inflammatory biomarkers in early symptomatic dengue are associated with progression to severe disease, and a hyperinflammatory phenotype in established severe dengue is associated with poor clinical outcomes, including death [10–13]. In addition to immunomodulation, HDT targeting endothelial hyperpermeability may have a role in ameliorating vascular leakage to reduce shock and respiratory distress.

The aims of this systematic mapping review were to characterise published and registered dengue therapeutic trials, describe their endpoints, and assess study design quality and internal validity to inform feasibility of meta-analysis. Our goal is to use the results to prioritise therapeutics for upcoming adequately powered platform trials evaluating both antiviral and host-directed treatments, and contribute to the development of a consensus core outcome set for use by the dengue trials community in future clinical trials [14].

## Methods

### Inclusion criteria

We included prospective clinical trials of dengue treatment which met the PICO criteria specified in **Table 1**. The studies could be randomised controlled trials, non-randomised trials or single-arm trials of all clinical phases. Registered but unpublished clinical trials were also included.

### Exclusion criteria

We excluded animal studies, trials of non-therapeutic interventions (e.g., diagnostics, vector-control, disease prevention studies etc.), pharmacokinetics and pharmacodynamics studies, human challenge studies and non-primary research studies, including post-hoc analysis, systematic reviews, meta-analyses, letters, opinions, editorials etc. We also excluded studies whose full-text or registration written in English could not be found.

### Search methods for identification of studies

An information specialist (EH) searched the following databases on 02/08/2023 and updated the search in full on 24/10/2024 using the search terms and strategy described in detail in S1 File Search strategies: Ovid MEDLINE, Ovid

**Table 1. PICO criteria.**

|  | Aim: review of therapeutic efficacy | Aim: review of endpoints |
|---|---|---|
| Participants | Patients with clinical or laboratory diagnosis of symptomatic dengue (at all clinical stages and of any age) | |
| Therapeutic Interventions | Included:<br>• Antiviral, i.e., drugs aiming to reduce dengue virus replication<br>• Host-directed therapy, i.e., drugs aiming to modulate the inflammatory/ immune response, vascular or other pathogenic pathways<br>• Other drugs proposed to modulate disease pathogenesis, without a specified mechanism<br>Excluded: Supportive therapies (no proposed mechanism of altering disease pathogenesis), e.g., fluid resuscitation, transfusion of blood products. | No restriction |
| Comparisons | No restriction (single-arm trials were also included) | |
| Outcomes | No restriction | |
| Types of studies | Published trials with full-text available | Published trials with full-text available, and trial registrations pending results. |

Embase, World Health Organisation International Clinical Trials Registry Platform (WHO ICTRP), and ClinicalTrials.gov. Results from both searches were incorporated in this report. Date of publication/registration was not restricted in this review. The search strategy was not restricted by the comparator to capture single-arm trials.

We imported the retrieved studies into excel sheets (for studies from WHO ICTRP) and Covidence (other databases) for screening. Pairs of assessors (TBH; TLT; NQH; LHBT and LMK) screened each study by title and abstract independently and retrieved full texts for potentially relevant trials, which were then reviewed by the same assessors in duplicate. We listed excluded studies and recorded reasons for exclusion. If new publications arising from the included trial registrations were identified during the review process, we incorporated the published full-text articles at the time of detection. We contacted trial authors for clarification where necessary and resolved any discrepancies through discussion with three of the other authors (NLV, JAW and SY).

## Data extraction and analysis

Data was extracted into a structured REDCap database. Three independent reviewers (TBH, TLT, NQH) extracted summary data on therapeutics used (antiviral or host-directed interventions), the day of illness, primary outcomes, study design, participant characteristics, recruitment methods, study country, date of study publication and study results if published. We did not extract secondary/exploratory outcomes. Two reviewers extracted data from a sample of eligible studies and achieved good agreement (above 80 percent), with the remainder extracted by one reviewer. TBH, LHBT and LMK performed data cleaning.

We used descriptive analyses to summarise the characteristics of all available trials on dengue, including the number of trials, treatments/arms, study populations, methods, main findings and the primary outcomes used. We chose not to carry out aggregate meta-analysis or effect estimation due to insufficient numbers of trials/patients for individual interventions, disparate endpoints, and concerns about methodological quality for many studies. Corticosteroids and *Carica papaya* leaf extract (CPLE, of variable regimens) were the most frequently studied interventions; given that recent meta-analyses already exist for these agents [15,16] and our search found no new high-quality trials to add, we did not perform an updated analysis.

## Risk of bias assessment

For studies evaluating the efficacy of proposed antiviral or HDT in dengue, we assessed the risk of bias using the revised Cochrane risk of bias tools: RoB-2 version 22Aug2019 [17] for randomised trials and ROBINS-I for non-randomised

studies of interventions; risk of bias assessments were primarily undertaken to systematically describe trial design rigour and assess feasibility for future data synthesis, not to support conclusions on treatment effect. We followed the Cochrane Handbook for Systematic Reviews of Interventions [18] to assess whether adequate steps had been taken to reduce the risk of bias across five domains: bias arising from the randomisation process, bias due to deviations from intended interventions, bias due to missing outcome data, bias in measurement of the outcome and in selection of the reported result. We categorised the risk of bias as 'high', 'some concerns', or 'low' for randomised trials and 'critical', 'serious', 'moderate' or 'low' for non-randomised studies of interventions. For the three most frequently evaluated interventions (corticosteroids, *Carica papaya* leaf extract and montelukast), two reviewers (of TBH, LMK and LHBT) independently assessed the risk of bias for each study; discrepancies were resolved through discussion with PKL. For the remaining 29 studies, one reviewer assessed the risk of bias. Another independent reviewer (NLV) randomly checked 25% of these assessments (8 studies) to detect discrepancies.

## Results

The search identified 7982 studies, of which 121 were included in this review; 72 were full-text publications and 49 were study registrations pending results, see Fig 1. The included trials were categorised into four groups based on the authors' proposed mechanism of action: antiviral (n = 10), HDT (n = 34), supportive (n = 31), and undefined (n = 46). The 31 studies which evaluated supportive treatment for dengue (mostly fluid regimens and platelet transfusion) were only reviewed for primary endpoints. The remaining 90 trials were reviewed in full.

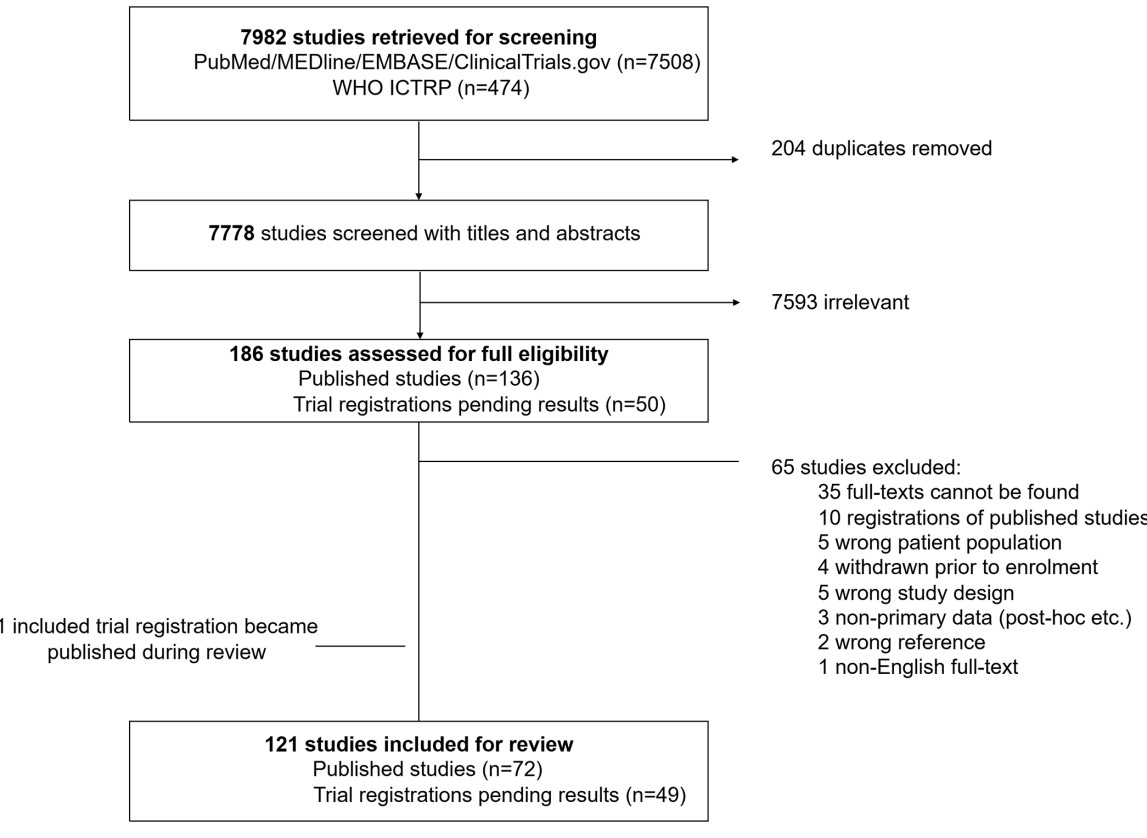

**Fig 1. PRISMA flow chart.**

## Overview of dengue therapeutic trials

Sample size of the included trials (the total number of participants enrolled in published studies and the planned sample sizes for unpublished trial registrations) are presented in Fig 2 - by country and region and in Fig 3 – by individual intervention. Most studies were conducted in South and Southeast Asia, with a few in Latin America. Only 11 of 72 (15.3%) published studies and 11 of 49 (22.4%) trial registrations had an actual or planned sample size of at least 250 participants.

After excluding supportive therapies, corticosteroids and *Carica papaya* leaf extract (CPLE) were the most frequently evaluated interventions, enrolling the largest number of participants (n = 944 and n = 920, respectively) and appearing in the greatest number of publications (n = 9 and n = 7, respectively). CPLE also had the highest number of trial registrations (n = 7), the majority from India, followed by vitamins C and D (n = 3 registrations each). The remaining 53 compounds were investigated in only one to three studies each.

With respect to age groups, children were the predominant participants in trials of supportive therapies and corticosteroids, but were under-represented in studies of other HDT, micronutrients and miscellaneous therapies (S1 Fig). No antiviral trials included paediatric participants.

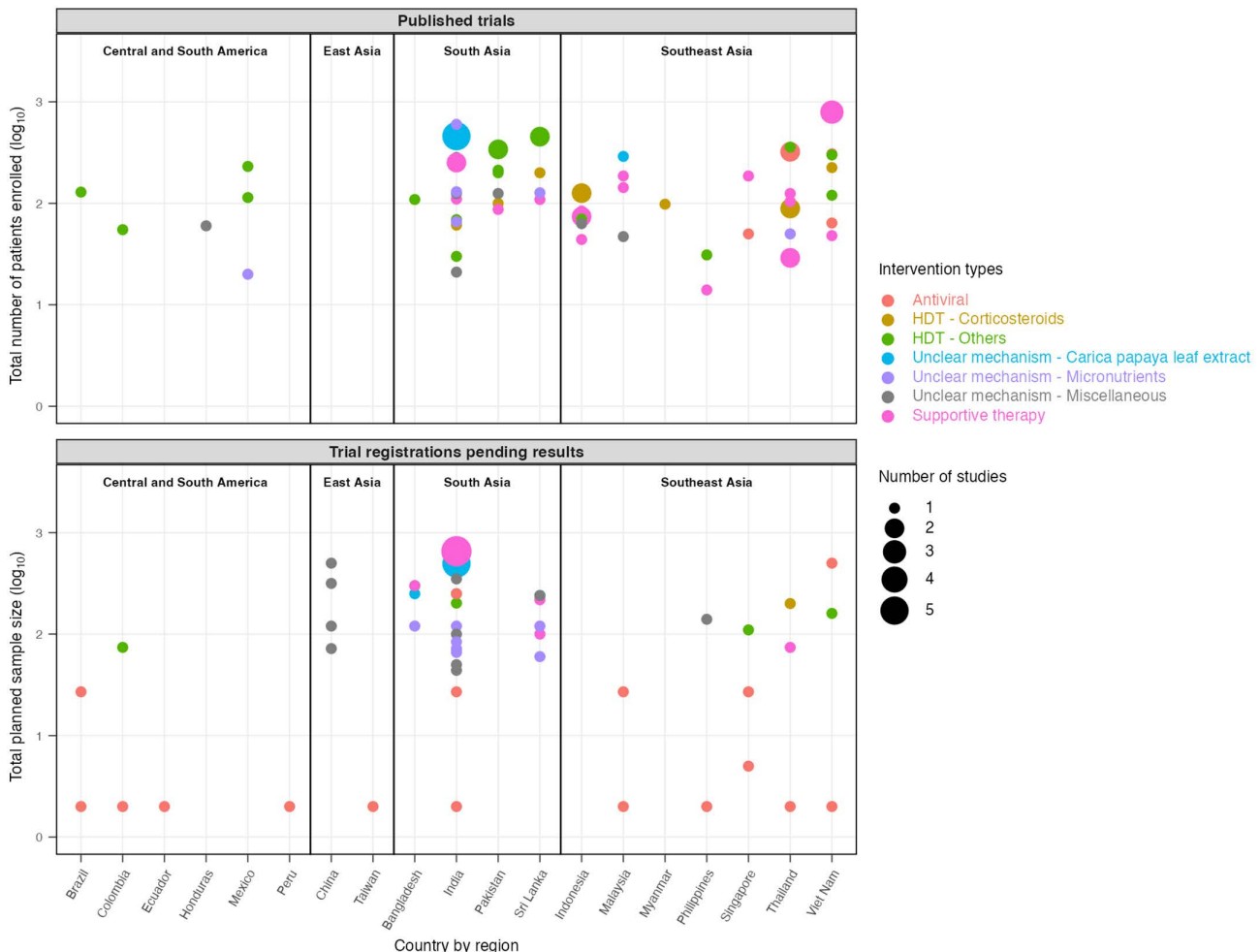

**Fig 2. Sample size of published and unpublished dengue therapeutic trials by country/region.**

**Fig 3. Sample size of published and unpublished dengue therapeutic trials by individual interventions.**

## Trials evaluating antivirals for dengue

We identified five published randomised, placebo-controlled clinical trials evaluating four repurposed agents with proposed antiviral activity against dengue: two trials evaluated the antiparasitic drug ivermectin [19], one evaluated chloroquine [20], and two investigated antiviral compounds originally developed for hepatitis C virus - celgosivir and balapiravir [21,22] (Table 2). None of these studies included participants under 15 years of age. No trial demonstrated a statistically significant difference in predefined virological or clinical outcomes between intervention and placebo groups. All but one of the studies were assessed as having a low risk of bias (Table 2 and S1 Table).

Review of WHO ICTRP and ClinicalTrials.gov identified five additional clinical trials that have either not yet concluded or have not published results. Two phase 2 trials - of AT-752, a nucleotide prodrug inhibitor of DENV polymerase (Clinicaltrials.gov: NCT05466240, Atea Pharmaceuticals, Inc) and JNJ-64281802, an NS3-NS4B inhibitor (Clinicaltrials.gov: NCT04906980, Janssen Research and Development) - were both terminated early due to recruitment challenges and sponsors' decisions to deprioritise dengue research. A phase 2 trial of EYU688, an NS4B inhibitor (Clinicaltrials.gov: NCT060066559, Novartis Pharmaceuticals) is ongoing. A phase 2 dose-ranging trial of Dengue Monoclonal antibody (Clinical Trials Registry India: CTRI/2021/07/035290, Serum Institute of India) has completed, with publication pending. An adaptive phase 2 trial evaluating

Table 2. Randomised controlled trials of interventions with proposed antiviral mechanism(s) of action.

| Study | Intervention(s) & Comparison(s) | Sample size | Country | Patient population | Confirmatory test | Primary endpoint(s) | Results | Overall risk of bias[*] |
|---|---|---|---|---|---|---|---|---|
| **Published studies** | | | | | | | | |
| Tricou 2010 | 1.Chloroquine 600mg at enrolment & day 2, 300mg on day 3<br>2.Matched placebo | 257◇ | Vietnam | Patients ≥ 15 y.o with dengue fever within 72 hours of onset | PCR/IgM | Time to viraemia clearance;<br>Time to NS1 clearance | No significant difference | + |
| Nguyen 2013 | Cohort 1:<br>1.Balapiravir 1500mg BID for 5 days<br>2.Matched placebo<br>Cohort 2:<br>1.Balapiravir 3000mg BID for 5 days<br>2.Matched placebo | 64 | Vietnam | Adult male patients 18–65 y.o with dengue fever within 48 hours of onset | NS1 | Safety and tolerability;<br>Viraemia AUC 0–168 hours;<br>Time to viraemia <1000 copies/mL;<br>Time to NS1 clearance | No major safety signals;<br>No measurable difference in virological endpoints | ! |
| Low 2014 | 1.Celgosivir 400mg within 6h of randomisation, then 200mg BID for 9 doses<br>2.Placebo | 50 | Singapore | Adults 21–65 y.o with dengue within 48 hours of onset | Dual PoC rapid test or PCR | Mean virological log reduction from baseline | No significant difference | + |
| Suputtamongkol 2021 | Phase II:<br>1.Ivermectin (IVM) 400 µg/kg/day for 2 days + placebo on 3rd day<br>2.IVM 400 µg/kg/day for 3 days<br>3.Placebo for 3 days | 108* | Thailand | Patients ≥ 15 y.o with dengue within 72 hours of fever onset | NS1 | Viraemia AUC 0–48 of 0–72 hours;<br>Time to viraemia and NS1 clearance | No significant difference | + |
| | Phase III:<br>1.IVM 400 µg/kg/day for 3 days<br>2.Placebo for 3 days | 203 | Thailand | | | Fever clearance;<br>Proportion of patients who developed DHF | No significant difference | + |
| **Trial registrations** | | | | | | | | |
| NCT06551844<br>(Not yet recruiting) | 1.Standard of care<br>2.Molnupiravir 800mg BID for 5 days<br>3.Dengue mAb 6mg/kg once only<br>New therapies may be added and poorly performing arms will be removed | NA | Vietnam;<br>other countries may be added | Patients ≥ 10 y.o with dengue fever within 48 hours of onset | NS1 | Viral clearance rate | NA | NA |

*(Continued)*

**Table 2.** (Continued)

| Study | Intervention(s) & Comparison(s) | Sample size | Country | Patient population | Confirmatory test | Primary endpoint(s) | Results | Overall risk of bias[*] |
|---|---|---|---|---|---|---|---|---|
| CTRI/2021/07/035290 (Completed) | 1. Dengue mAb 3 mg/kg IV over 2 hours; 2. Dengue mAb 5 mg/kg IV over 2 hours; 3. Dengue mAb 7 mg/kg IV over 2 hours; 4. Dengue mAb 9 mg/kg IV over 2 hours; 5. Placebo | 250 | India | Adults 18–60 y.o with dengue within 48 hours of fever onset. *Excluded:* severe dengue, significant lab abnormalities (e.g., low Hb, neutropenia, thrombocytopenia), known viral co-infections (HBV, HCV, HIV), substance abuse history | NS1 or PCR | Proportion of participants with related SAEs; Virological log reduction of infective dengue virus assessed by NS1 cell-based titration (NSET) assay 24hrs post treatment start | Not published | NA |
| NCT05466240 (terminated) | Cohort 1: 1.AT-752 750mg TID for 5 days; 2.Matched placebo. Cohort 2: 1.AT-752 dose A for 5 days; 2.Matched placebo. Cohort 3: 1.AT-752 dose B for 5 days; 2.Matched placebo | 21 | Brazil, Colombia, Ecuador, India, Malaysia, Peru, Philippines, Taiwan, Thailand, Vietnam | Adults 18–55 y.o with dengue fever within 48 hours of onset | NS1 or PCR | Change in dengue viral load from baseline | Terminated due to deprioritisation by sponsors and the impact of COVID-19 on recruitment | NA |
| NCT04906980 (terminated) | 1.JNJ-64281802: 2 initial loading doses up to day 2, followed by 1 maintenance dose/day on day 3–5; 2.Matched placebo | 5 | Singapore | Adults with dengue fever within 48 hours of onset | NS1 | $Log_{10}$-transformed viraemia AUC from day 1 to day 5 | No data reported for this outcome. Terminated due to reprioritisation of the Company's Communicable Diseases research [23] | NA |
| NCT06006559 (active) | 1.EYU688 (dose not specified); 2.Matched placebo | 108 | India, Malaysia, Singapore, Vietnam, Brazil | Adults with dengue fever within 48 hours of onset | NS1 or PCR | Viral load reduction on log scale 48 hours after starting treatment | NA | NA |

* 116 enrolled but 07 excluded to due undetectable baseline viraemia/NS1; 01 withdrawn.

◊ 307 patients enrolled but 50 did not have evidence of recent or acute dengue fever.

(*)Studies assessed as having a overall low risk of bias are shaded as green, some risk of bias as yellow and high or critical risk of bias as red.

Abbreviations: RCT, randomised controlled trial; AUC, area under the curve; NS1, non-structural protein 1; PoC, point of care; PCR, polymerase chain reaction; IgM, immunoglobulin M; DHF, dengue haemorrhagic fever; mAb, monoclonal antibody; PLT, platelet count; T, temperature; HBV, Hepatitis B virus; HCV, Hepatitis C virus; COVID-19, Coronavirus disease 2019; SAE, serious adverse events; IV, intravenous; PO, by mouth; OD, once daily; BID, twice daily; TID, three times daily; y.o, years old.

Dengue Monoclonal antibody and molnupiravir versus standard of care in early symptomatic dengue is registered but not yet recruiting (Clinicaltrials.gov: NCT06551844). All five registrations plan to enrol adult participants only.

## Trials evaluating host-directed therapy for dengue

We identified 34 trials of therapeutic agents with proposed host-directed mechanisms of action.

**Trials evaluating corticosteroids.** We identified nine completed trials evaluating oral or intravenous corticosteroid therapy in dengue of any severity, enrolling a total of 944 patients (329 with severe dengue, 290 with moderate disease, and 325 with early symptomatic dengue) (Table 3). The trials varied across several key parameters, including preparation, dose, duration, patient population (by age and disease severity), and primary outcomes. All were assessed as having either some concerns or a high risk of bias (Table 3 and S1 Table). An additional registered trial aims to evaluate dexamethasone in severe dengue (dose and duration not specified), but results have not been published.

Four randomised trials investigated the impact of corticosteroids on mortality in children with dengue shock, assessing different corticosteroid agents, doses and treatment durations. Sumarmo et al investigated a single dose of hydrocortisone [24], Tassniyom et al investigated a single dose of methylprednisolone [25] and two trials conducted in the 1970's [26,27] evaluated hydrocortisone for either three days or an unspecified duration. Each trial enrolled fewer than 100 paediatric patients, and none included adults. Three studies found no difference in mortality, while Min et al reported lower mortality among patients receiving hydrocortisone [26]. However, all four trials raised concerns regarding risk of bias due to unclear allocation methods, and open-label or unspecified blinding (Table 3 and S1 Table). Interpretation of the findings by Min et al is limited by the absence of baseline data on disease severity and key prognostic factors such as shock duration, comorbidities, and organ dysfunction.

Three randomised controlled trials have evaluated corticosteroids in adults with non-severe dengue and thrombocyto-paenia, with different doses and duration [28–30]. All measured changes in platelet count as the primary endpoint. None demonstrated a significant difference in platelet recovery between corticosteroid and placebo groups, except for one very small study assessed as having high risk of bias [30].

Two trials evaluated corticosteroids in early, uncomplicated dengue [31,32]. Tam et al evaluated low-dose and high-dose oral prednisolone administered within three days of illness onset, and found no significant difference in pre-defined clinical, haematological or virological outcomes (Table 3 and S1 Table). The remaining study was assessed as having high risk of bias.

Taken together, the available evidence is insufficient to draw conclusions regarding the impact of corticosteroid therapy on mortality in dengue shock, platelet recovery in dengue with thrombocytopaenia, or clinical and virological outcomes in early symptomatic dengue.

**Trials evaluating other host-directed therapeutics.** We identified 19 published studies evaluating a range of other host-directed therapies (HDTs) in patients with dengue. These included three trials of doxycycline; two each of montelukast, rupatadine and interleukin-11; and one each of carbazochrome sodium sulfonate (AC-17), intravenous immunoglobulin, colchicine, pentoxifylline, chloroquine, lovastatin, anti-D immunoglobulin, eltrombopag, oseltamivir and metformin. In addition, we identified five registered trials, each investigating zanamivir, anakinra, eltrombopag, ketotifen or doxycycline (Table 4).

Among therapies proposed to attenuate endothelial dysfunction, Malavige et al reported two trials of the platelet-activating factor receptor inhibitor rupatadine in patients with non-severe dengue; treatment did not significantly reduce the proportion developing pleural effusion, ascites or dengue haemorrhagic fever [33,34]. Whitehorn et al evaluated the safety of lovastatin in early symptomatic dengue; although the study was not powered for clinical outcomes, there were no differences in time to fever clearance or progression to severe disease between treatment and placebo groups [35]. Nitinai et al found that the leukotriene receptor antagonist montelukast did not reduce the incidence of dengue warning signs [36]. Tunjungputri et al reported that the sialidase inhibitor oseltamivir did not reduce plasma leakage, time to platelet recovery

**Table 3. Randomised trials of corticosteroids (proposed as host-directed therapy) for the treatment of dengue, subdivided by clinical phenotype.**

| Study | Intervention(s) & Comparison(s) | Sample size | Country | Patient population | Confirma-tory test | Primary end-point(s) | Results | Overall risk of bias(*) |
|---|---|---|---|---|---|---|---|---|
| **Published studies** | | | | | | | | |
| *Severe dengue* | | | | | | | | |
| Pongpanich 1973 | 1. Hydrocortisone IV 25mg/kg/day (duration not specified) 2. Standard of care | 71 | Thailand | Children 0.5 – 13 y.o with DSS | HI | Mortality | No deaths in either group | — |
| Min 1975 | 1. Hydrocortisone IV 25mg/kg on day 1, 15mg/kg on day 2, 10mg/kg on day 3 2. Standard of care | 98 | Burma | Children 0 – 8 + y.o with DSS | HI | Mortality | Significantly lower mortality in hydrocortisone group (18.75%) versus standard of care (44%) | ! |
| Sumarmo 1982 | 1. Hydrocortisone IV 50mg/kg once 2. Placebo | 97 | Thailand | Children ≤ 10 y.o with DSS | IgM | Mortality | No significant difference | ! |
| Tassniyom 1993 | 1. Methylprednisolone IV 30mg/kg once 2. Placebo | 63 | Thailand | Children ≤ 15 y.o with fluid refractory DSS | HIA, ELISA | Mortality | No significant difference | ! |
| *Non-severe dengue with thrombocytopenia* | | | | | | | | |
| Waly 1998 | 1. Methylprednisolone IV 1.2-1.5mg/kg/day for 5 days 2. Placebo (no further description available) | 29 | Indonesia | Adults ≥ 16 y.o with dengue and PLT < 100 G/L | IgM, IgG, HIA | PLT rise | No significant difference | — |
| Kularatne 2009 | 1. Dexamethasone IV 4mg, then 2mg TID for 24h 2. Placebo | 200 | Sri Lanka | Adults and children (12–65 y.o) with PLT < 50 G/L, no shock, no bleeding | IgM, HIA | PLT rise | No significant difference | — |
| Shashidhara 2013 | 1. Dexamethasone IV 8mg, then 4mg TID for 4 days 2. Placebo | 61 | India | Adults with dengue and PLT < 50 G/L, no shock, no bleeding | IgM | PLT rise | No significant difference | ! |
| *Early symptomatic dengue* | | | | | | | | |
| Tam 2012 | 1. Prednisolone PO 0.5mg/kg for 3 days 2. Prednisolone PO 2mg/kg for 3 days 3. Placebo | 225 | Vietnam | Children and young adults (5 – 20 y.o) with dengue fever ≤72 hours onset | NS1/IgM/PCR | Safety | No significant difference in duration of viraemia or side effects. | ! |
| Aslam 2021 | 1.PO fluids + PO cefixime + PO artemether/lumefantrine (AR/LR) 2.IV/PO fluids + PO cefixime + PO AR/LR 3.IV/PO fluids + PO cefixime + PO AR/LR + PO CPLE 4.IV/PO fluids + IV/PO cefixime + PO AR/LR + PO CPLE + IV dexamethasone | 100 | Pakistan | Adults 20 – 60 y.o with dengue fever | NS1 | PLT count | Significantly higher PLT count in groups receiving PO papaya extract; PLT count highest in group receiving PO CPLE and IV dexamethasone. | — |

*(Continued)*

**Table 3.** (Continued)

| Study | Intervention(s) & Comparison(s) | Sample size | Country | Patient population | Confirmatory test | Primary endpoint(s) | Results | Overall risk of bias(*) |
|---|---|---|---|---|---|---|---|---|
| **Trial registrations** | | | | | | | | |
| NCT05631405 (recruiting) | 1. Dexamethasone (dose unspecified)<br>2. Placebo | 200 | Thailand | Patients ≥ 7 y.o with severe dengue, E*xcluded:* severe disease > 24 hours, pregnancy, use of steroids in preceding 1 week | NS1/IgM | Mortality at 28 days | NA | NA |

**(\*)** Studies assessed as having a overall low risk of bias are shaded as green, some risk of bias as yellow and high or critical risk of bias as red.

Abbreviations: RCT, randomised controlled trial; NS1, non-structural protein 1; PoC, point of care; PCR, polymerase chain reaction; IgM, immunoglobulin M; IgG, immunoglobulin G; HIA, Haemagglutination inhibition assay; ELISA, enzyme-linked immunosorbent assay; DSS, dengue shock syndrome; PLT, platelet count; T, temperature; CPLE, carica papaya leaf extract; IV, intravenously; PO, by mouth; OD, once daily; BID, twice daily; TID, three times daily; y.o, years old.

or hospital discharge [37]. Salgado et al found that pentoxifylline infusion did not reduce mortality, duration of intensive care or hospital admission in children with DHF [38]. Tassniyom et al evaluated the proposed anti-permeability compound carbazochrome sodium sulfonate in children with DHF/DSS and found no difference in the proportion developing shock or pleural effusion between intervention and placebo groups [39]. We noted some concerns regarding risk of bias in the latter three publications.

Pannu et al reported that among Rhesus-positive patients with dengue and severe thrombocytopaenia, those who received anti-D immunoglobulin were more likely to achieve platelet counts above 50 G/L within 48 hours of the intervention than controls [40]. Chakraborty et al reported that a higher proportion of participants receiving a single dose of the thrombopoietin receptor antagonist eltrombopag had platelet counts above 150 G/L by day 7 after starting treatment, compared with the control group [41]. A phase III trial evaluating eltrombopag in 300 patients with dengue with warning signs is currently recruiting (SLCTR/2022/023). Nguyen et al reported that treatment with metformin had no effect on clinical or virological outcomes in overweight patients with dengue, and was poorly tolerated due to gastrointestinal side effects, making it unsuitable for further evaluation [42]. The remaining studies were assessed as being at high or critical risk of bias [43–50].

### Trials evaluating interventions with unclear mechanism(s) of action

Despite uncertain mechanisms of action, there has been considerable interest in *Carica papaya* leaf extract (CPLE), micronutrient supplementation, and other herbal preparations for dengue treatment. These accounted for 46 of the 89 studies identified. Table 5 summarises trials evaluating CPLE, while Table 6 presents studies of micronutrient supplementation and other interventions for which the proposed mechanism of action was not clearly specified by the investigators.

**Trials evaluating Carica papaya leaf extract.** We identified 14 studies evaluating CPLE, of which seven full-text articles were available, enrolling a total of 829 participants (Table 5 and S1 Table). The studies used varying formulations and extractions, including commercially produced Caripill tablets or syrup (Micro Labs, Bengaluru), and preparations made from fresh leaves, precluding comparison of dosing between studies. Measured outcomes related to platelet recovery for all studies.

**Table 4. Trials of other proposed host-directed therapies for the treatment of dengue.**

| Study | Intervention(s) & Comparison(s) | Sample size | Country | Patient population | Confirmatory test | Primary endpoints | Results | Overall risk of bias(*) |
|---|---|---|---|---|---|---|---|---|
| Published studies | | | | | | | | |
| *Trials evaluating doxycycline* | | | | | | | | |
| Castro 2011 | 1. Doxycycline PO for 7 days (children) or 10 days (adults) 2. Tetracycline PO for 7 days (children) or 10 days (adults) 3. Standard of care | 114 | Mexico | Patients 8–55 y.o with presumptive diagnosis of DHF or dengue fever and more than 2 days of illness | PCR | Serum cytokines and cytokine receptor/ antagonist levels | Significant decline in cytokine levels and increase in cytokine receptor/ antagonist levels in treatment groups compared to standard of care group. | — |
| Fredeking 2015 | 1. Doxycycline PO 200 mg initially then 100 mg BID for 7 days 2. Standard of care | 231 | Mexico | Adults with DHF (fever, haemorrhagic manifestations, PLT < 100 G/L and evidence of plasma leakage) | PCR | TNF and IL-6 levels and mortality | 46% lower mortality and significant decrease in TNF & IL-6 levels in doxycycline treatment group. | — |
| Kumar 2024 | 1. Doxycycline PO 100 mg BID for 5 days 2. Standard of care | 69 | India | Patients ≥ 12 y.o with dengue and warning signs or severe dengue *Excluded*: coinfection, fever >7days, pregnancy/lactation, oesophageal dysmotility | NS1, IgM, or PCR | Reduction in IL-6, TNF-α, ferritin and CRP | No significant difference | ! |
| *Trials evaluating montelukast* | | | | | | | | |
| Ahmad 2018 | 1. Montelukast PO 10mg OD for 5 days 2. Standard of care | 200 | Pakistan | Patients 13–65 y.o with dengue *Excluded* DSS at baseline, comorbidities | IgM, NS1 | Incidence of DSS | Significantly lower incidence of DSS in the Montelukast group. | — |
| Nitinai 2024 | 1. Montelukast PO 10mg OD for 10 days or until resolved/ convalescence rash developed 2. Placebo | 358 | Thailand | Adults ≥ 20 y.o with dengue *Excluded* any warning signs, pregnancy, critical illness or concurrent diagnoses. | NS1 or PCR | Incidence of dengue with warning signs | No significant difference | + |
| *Trials evaluating rupatadine* | | | | | | | | |
| Malavige 2018 | 1. Rupatadine PO 40mg for up to 5 days (or until discharge) 2. Placebo | 183 | Sri Lanka | Adults 18–60 y.o with dengue within 5 days of fever onset *Excluded:* evidence of vascular leak | NS1 | Incidence of fluid leakage (pleural effusions, ascites) | No significant difference | + |
| Malavige 2022 | 1. Rupatadine PO 40mg OD for 5 days 2. Placebo | 271 | Sri Lanka | Adults 18–60 y.o with dengue within 3 days of fever onset *Excluded*: pregnancy, alcohol/ drug dependence, hepatic or renal failure. | NS1, PCR | Incidence of DHF (fluid leakage) | No significant difference | + |

*(Continued)*

| Study | Intervention(s) & Comparison(s) | Sample size | Country | Patient population | Confirmatory test | Primary endpoints | Results | Overall risk of bias[*] |
|---|---|---|---|---|---|---|---|---|
| *Trials evaluating interleukin-11* | | | | | | | | |
| Suliman 2014 | 1. Recombinant human IL-11 SC 1.5 mg once 2. Placebo | 40 | Pakistan | Adults <50 y.o with dengue and PLT > 10G/L *Excluded:* active bleeding, need for transfusion of any blood product | IgM | PLT rise | Clinically marginal but statistically significant improvement in PLT count in IL-11 treatment group. | — |
| Khan 2024 | (Non-randomised trial) 1. Human interleukin-11 analog SC 1.5mg/day for 5 days 2. Standard of care | 300 | Pakistan | Adults 18–65 y.o with dengue and PLT < 30G/L *Excluded:* comorbidities | NS1, IgM/IgG ELISA | Not specified. Authors focused on PLT count. | Significant increase in PLT count in IL-11 treatment group. | — (ROBINS-I) Critical risk of bias |
| *Trials evaluating other host-directed therapies* | | | | | | | | |
| Tassniyom 1997 | 1. Carbazochrome sodium sulfonate (AC-17) IV 25mg bolus, then continuous infusion for 3 days 2. Vitamin B as Placebo | 95 | Thailand | Children ≤ 14 y.o with a presumptive diagnosis of DHF/DSS | IgM, HI | Plasma leakage DSS | No significant difference | ! |
| Dimaano 2007 | 1. IV immunoglobulin 0.4g/kg/day for 3 days 2. Standard of care | 31 | Philippines | Non-primary dengue within 5 days of illness and PLT 20–80 G/L *Excluded* prominent bleeding or shock | IgM | PLT rise | No significant difference | — |
| Salgado 2012 | 1. Pentoxifylline IV 12.5mg/kg/day for 3 days. 2. Standard of care | 55 | Colombia | Children with dengue within 5 days of illness onset | NS1, IgM | ICU admission, hospital stay and serum TNF-α | No significant difference in length of hospital or ICU stay. Significantly lower TNF-α at 24 hours in pentoxifylline treatment group. | ! |
| Qazi 2012 | 1. Colchicine PO 0.5 mg BID for 72 hours 2. Sodium bicarbonate PO BID for 72 hours | 212 | Pakistan | Patients 14–70 y.o with probable dengue fever and PLT 20–100 G/L *Excluded*: DHF/any warning signs, serious mucosal bleeding, hypotension (systolic blood pressure <90mmHg or diastolic blood pressure <60mmHg), known platelet disorder | None | PLT rise | No significant difference | — |

*(Continued)*

| Study | Intervention(s) & Comparison(s) | Sample size | Country | Patient population | Confirmatory test | Primary endpoints | Results | Overall risk of bias[*] |
|---|---|---|---|---|---|---|---|---|
| Borges 2013 | 1. Chloroquine PO 500mg BID for 3 days<br>2. Placebo | 37◊ | Brazil | Adults with dengue-related symptoms for less than 72 hours | PCR, IgM, NS1 (≥2/3) | Duration of disease | No significant difference | — |
| Whitehorn 2016 | 1. Lovastatin 80mg OD for 5 days<br>2. Placebo | 300 | Vietnam | Adults ≥ 18 y.o with dengue within 72 hours of fever onset *Excluded*: ALT > 150U/L, CK > 1000U/L, PLT < 50G/L, pregnancy/ lactation, cirrhosis or myopathy, current use of statins | NS1, PCR | Safety (adverse events, serious adverse events) | No significant difference in safety events, or secondary outcomes. | + |
| Pannu 2017 | 1. Anti-D IV 50 μg/kg (250 IU/kg) once<br>2. Standard of care | 30 | India | Adults 18–65 y.o with dengue and PLT ≤ 20 G/L, Rh+ *Excluded* comorbidities | NS1, IgM | Proportion of patients with PLT > 50G/L | Significantly higher proportion with PLT > 50 G/L at 48 hours in anti-D treatment group (60%) versus standard of care (6.7%) | ! |
| Chakraborty 2020 | 1. Eltrombopag PO 25mg OD for 3 days<br>2. Eltrombopag PO 50mg OD for 3 days<br>3. Standard of care | 109 | Bangladesh | Patients 15–65 y.o with dengue with ≥ 2 warning signs or severe dengue and PLT < 100 G/L *Excluded:* comorbidities, AST or ALT > 5ULN, portal vein thrombosis, HBV/HCV infection. | NS1, IgM/IgG | % patients with PLT > 150G/L | Significantly higher proportion of patients with PLT ≥ 150 G/L in eltrombopag treatment groups (91%) at day 7 versus standard of care (55%). | ! |
| Tunjungputri 2022 | 1. Oseltamivir phosphate PO 75mg BID for 5 days (or until discharge)<br>2. Placebo | 70 | Indonesia | Adults ≥ 16 y.o with dengue within 6 days of fever onset and PLT < 70 G/L *Excluded* AST or ALT > 3xULN or clinically significant bleeding | NS1, serology | Time to PLT recovery (>100g/L) and plasma leakage (gallbladder thickness, HCT, albumin, syndecan-1) | No significant difference in time to PLT recovery or parameters of plasma leakage | ! |
| Nguyen 2025 | Cohort 1<br>1. Metformin PO 850mg (adults) or 500mg (children) OD for 5 days<br>2. Standard of care<br>Cohort 2<br>1. Metformin PO 1500mg/day (>60 kg) or 500mg BID (<60 kg) for 5 days<br>2. Standard of care | 120 | Vietnam | Patients aged 10–30 y.o. with dengue within 72 hours of fever onset and BMI ≥ 25 *Excluded*: severe dengue | NS1 | Safety and tolerability | Metformin was poorly tolerated. No significant difference in secondary outcomes. | + (ROBINS-I) |

*(Continued)*

| Study | Intervention(s) & Comparison(s) | Sample size | Country | Patient population | Confirmatory test | Primary endpoints | Results | Overall risk of bias[*] |
|---|---|---|---|---|---|---|---|---|
| Trial registrations | | | | | | | | |
| NCT04597437 | 1. Zanamivir 600mg (≥50 kg) or 12mg/kg (<50 kg) IV BID for days 2. Placebo | 74 planned | Colombia | Patients ≥7 y.o with dengue with warning signs AND fever ≥38 in the last 24 hours | NS1 | Incidence of treatment-emergent Adverse Events | NA | NA |
| NCT05611710 | 1. Anakinra IV 200mg BID for 4 days (2mg/kg for patients <16 y.o who weigh <50 kg) 2. Placebo | 160 planned | Vietnam | Patients ≥ 12 y.o with clinical diagnosis of dengue with warning sign(s) or severe dengue, and ferritin >2000µg/L | Clinical diagnosis | Change in mSOFA score over 4 days | NA | NA |
| NCT02673840 | 1. Ketotifen 2mg BID for 10 doses 2. Placebo | 112 planned | Singapore | Adults 21–60 y.o with laboratory-confirmed dengue within 72 hours of illness onset *Excluded:* "symptoms of severe dengue such as persistent vomiting, liver enlargement >2cm, AST or ALT>1000U/L", renal impairment, comorbidities | NS1, PCR | Reduction of pleural effusion by MRI | NA | NA |
| SLCTR/2022/023 | 1 - Eltrombopag 25mg OD for 3 days 2 - Placebo | 300 planned | Sri Lanka | Adults 18–65 y.o with laboratory-confirmed dengue, any warning sign and PLT < 100G/L *Excluded*: Pregnancy, severe dengue, ALT > 5ULN, comorbidities, pleural effusion, other PLT disorders, portal vein thrombosis, hepatitis (HCV/HBV), immunosuppressive therapy. | NS1/IgM | PLT count Proportion of patients with PLT > 150G/L | NA | NA |

*(Continued)*

**Table 4.** (Continued)

| Study | Intervention(s) & Comparison(s) | Sample size | Country | Patient population | Confirmatory test | Primary endpoints | Results | Overall risk of bias[(*)] |
|---|---|---|---|---|---|---|---|---|
| CTRI/2018/01/011548 | 1 - Doxycycline 5mg/kg BID for 7 days<br>2 - Placebo | 202 planned | India | Children 8–18 y.o with confirmed dengue<br>*Excluded*: Unable to take oral medications, hypersensitivity to doxycycline, on antiepileptic drugs, or already received doxycycline for current illness. | Not specified | Mortality | NA | NA |

◊129 randomised but only 37 confirmed dengue and agreed to participate.

(*) Studies assessed as having a overall low risk of bias are shaded as green, some risk of bias as yellow and high or critical risk of bias as red.

Abbreviations: RCT, randomised controlled trial; NS1, non-structural protein 1; PoC, point of care; (RT-)PCR, (real time) polymerase chain reaction; IgM, immunoglobulin M; IgG, immunoglobulin G; ELISA, enzyme-linked immunosorbent assay; DSS, dengue shock syndrome; DHF, dengue haemorrhagic fever; PLT, platelet count; ALT, alanine aminotransferase; AST, aspartate aminotransferase; ULN, upper limit of normal; T, temperature; BMI, body mass index; TNF, tumour necrosis factor; TNF-α, tumour necrosis factor alpha; IL-6, interleukin 6; CRP, c-reactive protein; mSOFA, modified sequential organ failure assessment; HBV, hepatitis B virus; HCV, hepatitis C virus; IV, intravenously; PO, by mouth; OD, once daily; BID, twice daily; TID, three times daily; y.o, years old.

Subenthiran et al, the largest published trial (n = 290), reported higher platelet counts at 48 hours post-intervention in the CPLE group compared with controls [51]. However, the magnitude of this difference was small (69.5 vs 79.6 G/L at 48 hours) and we had some concerns regarding risk of bias from missing outcome data in the publication. The remaining six publications also reported efficacy measured by platelet count improvement, however, they were either very small trials or were assessed as being at high or critical risk of bias [52–57]. Taken together, there is insufficient high-quality evidence that CPLE improves clinical outcomes in dengue.

We also identified seven unpublished trials registered between 2014 and 2019 evaluating CPLE. Most examined platelet recovery as the primary endpoint, while one aimed to assess plasma leakage (SLCTR/2017/034). We were unable to confirm the current status of these trials.

**Trials evaluating micronutrient supplementation.** We identified six published studies evaluating oral micronutrient supplementation in patients with dengue: three evaluated vitamin E, two Vitamin C, and one each zinc and calcium carbonate (Table 6). The proposed mechanisms of action, and rationale for the selected primary outcomes were not clearly described. Among the vitamin E trials, both Mittal et al and Vaish et al reported higher platelet counts in participants receiving vitamin E compared to controls, but we had some concerns regarding risk of bias for both studies [58,59]. Mittal et al also reported higher platelet counts in participants receiving vitamin C supplementation versus placebo; however, the vitamin doses were not specified. Rerksuppaphol et al, whose trial was assessed as having low risk of bias, found no difference in time to defervescence between children with dengue who received zinc for 5 days and those given placebo [60]. The remaining studies were assessed as being at high or critical risk of bias [61–63] (Table 6 and S1 Table). Overall, there is insufficient evidence to draw conclusions on the efficacy of vitamin E, vitamin C, zinc or calcium carbonate supplementation in dengue virus infection.

**Miscellaneous studies.** Seven publications evaluated miscellaneous or unspecified compounds (Table 6 and S1 Table). We had some concerns regarding risk of bias in one study evaluating Propoelix [64]; the remaining studies were

**Table 5. Trials of Carica papaya leaf extract for the treatment of dengue.**

| Study | Intervention(s) & Comparison(s) | Sample size | Country | Patient population | Confirmatory test | Primary endpoint(s) | Results | Overall risk of bias (*) |
|---|---|---|---|---|---|---|---|---|
| **Published studies** | | | | | | | | |
| Sathyapalan 2020 | 1. CPLE (Caripill) tablets 1100mg TID for 5 days 2. Placebo | 50 | India | Adults ≥18 y.o with dengue and PLT<30 G/L *Excluded*: pregnancy, lactation, on steroids/other indigenous meds, recent PLT transfusion, ALT>150 U/L, CK>1000 U/L and myopathy. | NS1/IgM | PLT & HCT change | Higher mean % increase in PLT count at 72 hours in CPLE group. No difference in HCT. | ! |
| Perumal 2019 | 1. CPLE (ZYBORICA) PO 1100mg TID for 5 days 2. Placebo | 60 | India | Adults 18–60 y.o with DF/DHF grade I-II and PLT 30 – 150 G/L *Excluded*: pregnancy/lactation, DHF grade 3–4, ALT>165U/L, PLT<30G/L, any blood products during current illness, haematological disorders, impaired renal function, use of an investigational drug within preceding 1 month. | NS1 | PLT count | Higher PLT count in CPLE group from day 3 – 6 | – |
| Srikanth 2019 | 1. CPLE (Caripill) 275mg if 1–5 years (550mg if >5 years) TID for 5 days 2. Standard of care | 294 | India | Children 1–12 y.o with DF/DHF grade I-II and PLT 30–100 G/L *Excluded*: DHF grade 3–4, PLT<30 G/L, transfusion of blood products, renal impairment, AST or ALT>3ULN. | NS1 | PLT rise | Significantly higher PLT count on day 5 of study in CPLE group | – |
| Cordeiro 2019 | 1. CPLE (DEN-PAP) 15ml BID for 05 days | 15 (Single arm) | India | Adults 18–60 y.o with dengue and low PLT but *not<70G/L* *Excluded:* normal PLT count, pregnancy/lactation; bleeding; any co-infection; transfusion of PLT/whole blood <120 days | IgM/IgG | PLT rise | Authors reported a 'significantly higher PLT count with treatment' (but mean PLT was 120 G/L at baseline and below 140 G/L after 5 days), no statistical tests. | Critical risk of bias (ROBINS-I) |
| Srikrishna 2018 | 1. Papaya, Gudichi and Maricha (Platenza) 2 tablets PO BID for 10 days 2. CPLE 1 tablet PO TID for 10 days (dose not specified) | 40 | India | Adults 18–60 y.o with DF/DHF grade I-II and PLT 30-100G/L *Excluded:* PLT<30G/L; DHF grade 3–4; hypotension/hypovolemia; any skin/mucosal bleeding; haematological disorders or taking blood thinning medications; use of any blood products during preceding 1 month; pregnancy/lactation | IgG, IgM and NS1 | PLT rise | Higher PLT count on day 6 and 8 in Platenza group compared to control group. | – |

*(Continued)*

| Study | Intervention(s) & Comparison(s) | Sample size | Country | Patient population | Confirmatory test | Primary endpoint(s) | Results | Overall risk of bias (*) |
|---|---|---|---|---|---|---|---|---|
| Subenthiran 2013 | 1. CPLE (30ml juice extracted from 50g fresh leaves) OD for 3 days 2. Standard of care | 290 (38 LFU, 24 missing data) | Malaysia | Adults 18–60 y.o with laboratory-confirmed DF/DHF grade I-II and PLT < 100 G/L *Excluded*: DHF grade 3–4, pregnancy, lactation, transfusion of blood products, CK > 500 U/L, AST or ALT > 3xULN. | NS1/IgM | PLT & HCT change | Higher mean PLT count at 40 hours and 48 hours in CPLE group compared to control group. | (yellow) |
| Yunita 2012 | 1. CPLE capsules 550mg containing 70% ethanol extract of *C. papaya* leaves (frequency & duration unknown) 2. Standard of care | 80 | Indonesia | Patients 15–55 y.o with clinical dengue on days 2–7 of illness, PLT < 150 G/L and >20% rise in HCT *Excluded*: haematological disorders, severe bleeding. | None | PLT & HCT change | Significant increase in PLT by day 5 in CPLE groups compared to control group. | – (red) |
| **Trial registrations pending results** | | | | | | | | |
| CTRI/2019/04/018435 | 1 - CPLE PO 1100 mg TID for 5 days 2 - Placebo (multivitamins) | 176 planned | India | Adults (>18 years) admitted dengue fever *Excluded*: Chronic liver disease, co-infections (e.g., malaria, typhoid, scrub typhus), PLT disorders (e.g., idiopathic purpura thrombocytopenia), or pregnancy. | NS1/IgM | PLT count | NA | NA |
| CTRI/2018/08/015526 | 1 - CPLE (CARIMAX) PO 1100mg TID for 5 days 2 - Placebo | 60 planned | India | Adults 18–60 years with DF or DHF grade I–II *Excluded*: Pregnancy/lactation, DHF grade III–IV, PLT < 20G/L, active bleeding, blood transfusion during current illness, haematologic disorders, ALT > 165 U/L, impaired renal function, investigational drug use within 1 month | Unspecified | PLT count | NA | NA |
| SLCTR/2017/034 | 1 – CPLE (from 50g of fresh papaya leaves) PO 25ml BID for 5 days 2 – Standard of care | 240 planned | Sri Lanka | Patients 14–70 y.o with dengue and PLT < 150G/L *Excluded*: Plasma leakage on ultrasound, pregnancy/lactation, decompensated cardiac failure, renal/hepatic disease, body mass index >35 kg/m², allergy to papaya, history of bleeding/platelet disorders, active peptic ulcer, anticoagulant use. | NS1/IgM | Occurrence and severity of plasma leakage | NA | NA |

*(Continued)*

| Study | Intervention(s) & Comparison(s) | Sample size | Country | Patient population | Confirmatory test | Primary endpoint(s) | Results | Overall risk of bias (*) |
|---|---|---|---|---|---|---|---|---|
| CTRI/2017/08/009579 | 1 – CPLE (Caripill) PO 1100mg TID for 5 days<br>2 – Placebo | 100 planned | India | Adults ≥18 y.o with dengue and PLT ≤ 30G/L<br>*Excluded*: Pregnancy/lactation, prior steroid use or PLT transfusion, use of indigenous medicines, or CK > 1000 U/L. | NS1 ELISA and/or PCR | PLT & HCT; Use of blood products | NA | NA |
| CTRI/2017/07/009139 | 1 – CPLE (Caripill) PO 1100 mg TID for 5 days<br>2 – Placebo | 100 planned | India | Adults 15–70 y.o with dengue and PLT 10 – 100 G/L<br>*Excluded*: Pregnancy/lactation, DHF grade III–IV, active bleeding, recent blood transfusion, haematologic disorders, use of papaya leaf extract, use of investigational drug within 1 month. | Ag/Ab (unspecified) | PLT count | NA | NA |
| CTRI/2014/10/005120 | 1 - CPLE (PAPA-YUR) PO 600mg BID for 12 days<br>2 - Standard of care | 60 planned | India | Adults 18–60 y.o with dengue<br>*Excluded*: bleeding beyond petechiae/purpura, pregnancy/lactation, known hepatic/renal disease, inherited PLT disorders, NSAID use, papaya allergy, or >2 vomiting episodes in past 24 hours. | NS1, IgM/IgG or PCR | PLT aggregation, membrane fluidity and p-selectin expression | NA | NA |
| NCT06121934 (completed) | 1. CPLE (Caripill) tablets 1000mg TID 5 days<br>2. Placebo | 300 planned | Bangladesh | Adults with dengue<br>*Excluded:* pregnancy or lactation, steroids/immunosuppressant, PLT transfusion, other causes of thrombocytopenia, ALT > 150U/L, serum creatinine >1.4mg/dl, CK > 1000U/L | NS1/IgM | PLT & HCT change | NA | NA |

(*) Studies assessed as having a overall low risk of bias are shaded as green, some risk of bias as yellow and high or critical risk of bias as red.

Abbreviations: ALT, alanine aminotransferase; AST, aspartate aminotransferase; BID, twice daily; BMI, body mass index; CK, creatinine kinase; CPLE, Carica papaya leaf extract; DF, dengue fever; DHF, dengue haemorrhagic fever; ELISA, enzyme-linked immunosorbent assay; HCT, haematocrit; IgG, immunoglobulin G; IgM, immunoglobulin M; IV, intravenously; NA, not available; NS1, non-structural protein 1; NSAID, nonsteroidal anti-inflammatory drug; OD, once daily; PLT, platelet count; PO, by mouth; (RT-)PCR, (real time) polymerase chain reaction; T, temperature; TID, three times daily; ULN, upper limit of normal; y.o, years old.

assessed as being at high or critical risk of bias [65–70]. There is no high-quality evidence that any of these compounds provide clinical benefit in dengue virus infection.

### Patient endpoints

The primary endpoints used in dengue therapeutic trials showed high heterogeneity (Figs 4–5).

**Primary endpoints used in trials of antiviral treatments.** Among interventions with a proposed antiviral mechanism of action (Table 2, n = 10), most studies (8/10) used virological outcomes based on viral kinetics as their primary endpoints

**Table 6. Trials of other therapeutic interventions (excluding trials of symptomatic treatments).**

| Study | Intervention(s) & Comparison(s) | Sample size | Country | Patient population | Confirmatory test | Primary endpoint(s) | Results | Overall risk of bias (*) |
|---|---|---|---|---|---|---|---|---|
| **Published trials** | | | | | | | | |
| *Micronutrients* | | | | | | | | |
| Cabrera-Cortina 2008 | 1. Calcium carbonate for 1 week (400–600mg/d first 2–3 days, then 200–400mg/d) 2. Standard of care | 20 | Mexico | Clinical diagnosis of DF and tourniquet test positive within 3–5 days of fever onset *Excluded:* "alarm signs" (severe abdominal pain, prolonged vomiting, abrupt change from fever to hypothermia, and changes in consciousness) | Not specified | PLT rise | Significantly higher PLT count in intervention group compared to placebo. | – (Critical, ROBINS-I) |
| Vaish 2012 | 1. Vitamin E 400mg PO OD, duration not specified 2. Standard of care | 66 | India | Patients with DF and PLT 10–100 G/L *Excluded:* Any PLT transfusion during present illness; overt bleeding; use of antiplatelet medications; family history of bleeding diathesis; severe illness | Clinical/ NS1/IgM | PLT rise PLT transfusion requirement | Higher mean PLT count on day 4 and day 7 and lower proportion requiring PLT transfusion in vitamin E group compared to control group. | ! |
| Chathurangana 2017 | 1. Vitamin E 200–400mg PO OD for up to 7 days 2. Placebo | 127 | Sri Lanka | Children 5–12 y.o with clinical diagnosis of DF within 84 hours of fever onset *Excluded:* chronic illness or long-term medication use. | NS1/IgM/ IgG | Unspecified | No difference in the duration of hospital stay or occurrence of leaking but shorter duration of leaking in the treatment group. | – |
| Rammohan 2018 | 1. Vitamin C 500mg PO QID 2. Standard of care | 130 | India | Adults with DF and PLT 50–100 G/L *Excluded*: Bleeding complications; requiring platelet transfusion; use of iron therapy, anticoagulants/steroids/aspirin; use of high dose Vitamin C in the last 1 month; abnormal renal function tests; pregnancy and lactation | NS1 | Mean PLT count | Paper reported significant change between 2 groups but not direction of change. | – |
| Rerksuppaphol 2018 | 1. Zinc (15mg elemental) PO TID for 5 days 2. Placebo | 50 | Thailand | Children ≥ 1 y.o with DF/ DWS *Excluded*: Regular use of vitamins or minerals; chronic systemic diseases | NS1/IgM/ IgG | Recovery time based on day of convalescence of fever | No significant difference in mean time to defervescence | + |

*(Continued)*

| Study | Intervention(s) & Comparison(s) | Sample size | Country | Patient population | Confirmatory test | Primary endpoint(s) | Results | Overall risk of bias (*) |
|---|---|---|---|---|---|---|---|---|
| Mittal 2024 | 1. Vitamin E PO OD, dose unspecified 2. Vitamin C PO OD, dose unspecified 3. Placebo | 600 | India | Adults ≥18 y.o with DF and PLT<100G/L *Excluded*: Pregnancy/lactation, vitamin C/E hypersensitivity, severe comorbidities or need for intensive care | NS1/PCR | Change in PLT count | Vitamin E and vitamin C significantly increased platelet counts compared to the placebo group. | ! |
| *Miscellaneous* | | | | | | | | |
| Jacobs 2007 | 1. Homeopathy (variable regimens, not specified) 2. Placebo | 60 | Honduras | Patients ≥ 12 y.o with probable DF ≤72 hours *Excluded:* History of anaemia, malaria, liver disease, or arthritis | IgM/HI/virological culture | Duration of symptoms (fever, pain) | No significant difference. Only 3 subjects had laboratory confirmed dengue. | – |
| Mir 2012 | Euphorbia hirta linn (or Tawa- tawa), dose and regimen not specified | 125 | Pakistan | Confirmed dengue | Not specified | Blood parameters (PLT, HCT, WBC) | No significant change. | – (ROBINS-I) |
| Soroy 2014 | 1. Propoelix 200mg TID for 7 days 2. Placebo | 63 | Indonesia | Patients 18–45 y.o with DHF (WHO 1997) *Excluded*: DSS, PLT<20 G/L, allergy to bee products, any comorbidities | IgM/IgG | Impact on the clinical course | Propoelix group had higher mean PLT count by day 6–7, and a shorter hospital stay versus placebo group | ! |
| Nayak 2019 | 1. "Homeopathy" (variable regimens, not specified) 2. Standard of care | 283 | India | DF with thrombocytopenia | Clinical (probable)/IgM/NS1 | Change in PLT count | Higher mean increase in PLT count from day 1 to day 5 of follow-up in the homeopathy group compared to the standard of care group. | – (ROBINS-I) |
| Muhamad 2020 | 1. Kelulut honey (Trigona) PO 3.5g BID for 3 days 2. Corn syrup as placebo | 47 | Malaysia | Adults >18 y.o with dengue fever | Serology (unspecified) | Clinical warning signs (not specified) | No improvement in clinical warning signs or blood parameters (HCT, WBC) | – |
| Kumar 2022 | 1. NilavembuKudineer (NKN) + Honey water PO BID for 5 days 2. Placebo (Honey water) | 21 | India | Adults 18–60 y.o with dengue *Excluded:* comorbid infections, chronic diseases, pregnancy/lactation, transfusion of any blood products, haematological disorders, or medications affecting platelets (e.g., NSAIDs, anticonvulsants). | Unspecified | Antipyretic effect PLT count | Significant decrease in body temperature after 5 days in both groups. Higher decrease in NKN group. | – |

*(Continued)*

**Table 6.** (Continued)

| Study | Intervention(s) & Comparison(s) | Sample size | Country | Patient population | Confirmatory test | Primary endpoint(s) | Results | Overall risk of bias (*) |
|---|---|---|---|---|---|---|---|---|
| Pawar 2024 | 1. Guduchi Kwath 2. Kiratatikta Kwath 3. 'Natural Group' | 120 | India | Adults 18–70 y.o with dengue fever and thrombocytopenia *Excluded:* PLT<20G/L; other infections; history of any bleeding disorder; DSS, alcoholism/ liver cirrhosis; dengue encephalitis; GI bleeding; lactation, menstruation; aspirin and NSAIDs; stroke; myocardial infarction; Chikungunya | NS1/IgM/ IgG | Not specified in publication | Higher PLT count in Guduchi Kwath group compared to 'natural' group at day 7 (232 vs 180 G/L) | _ (ROBINS-I) |
| **Trial registrations** | | | | | | | | |
| *Micronutrients* | | | | | | | | |
| NCT05034809 | 1. Melatonin 20mg PO OD for 5 days 2. Standard of care | 140 | Philippines | Children 5–18 y.o with DF/DWS *Excluded*: Severe dengue/DSS, nil by mouth, previously treated in a referring facility, poor tablet tolerance | NS1 (DF) Clinical (DWS) | PLT rise | Not available | NA |
| NCT06071481 (recruiting) | 1. Vitamin D 2,00,000 IU PO 2. Vitamin D 4,00,000 IU PO 3. Standard of care | 120 | Bangladesh | Adults 18–65 y.o with NS1-positive dengue within 72 hours of fever onset *Excluded*: critical illness, pregnancy, Vit D hypersensitivity, high Ca2+, hypoalbuminemia, malignancy, nephrolithiasis, renal impairment | NS1 | Proportion of mortality and progression to severe dengue | Not available | NA |
| CTRI/2024/05/ 066756 | 1. Calcium carbonate 500mg BID for 15 days 2. Vitamin D 4000IU OD for 15 days 3. Placebo | 120 | India | Adults 18–60 y.o with confirmed dengue *Excluded*: pregnancy, DSS/DHF, any comorbidities, allergy to drug | Unspecified | Progression to DSS and DHF | Not available | NA |
| CTRI/2023/10/ 059167 | 1. Vitamin D 2000IU PO OD for 10 days 2. Standard of care | 72 | India | Adults 18–60 y.o with dengue *Excluded*: Comorbidities (except diabetes/ hypertension), co-infections, coagulation disorders, antiplatelet/ anticoagulant use, transfusion, CKD, chronic liver disease, pregnancy/ lactation. | NS1 | Progression to severe dengue | Not available | NA |

*(Continued)*

| Study | Intervention(s) & Comparison(s) | Sample size | Country | Patient population | Confirma-tory test | Primary end-point(s) | Results | Overall risk of bias (*) |
|---|---|---|---|---|---|---|---|---|
| SLCTR/2023/007 (recruiting) | 1. Vitamin E PO 200–400mg OD (age dependent) until day 7 of illness 2. Standard of care | 120 | Sri Lanka | Children 5–14 y.o with dengue within 84 hours of fever onset *Excluded*: Chronic illness, long-term medications, inability to take medication | NS1 | Occurrence and duration of plasma leakage | Not available | NA |
| CTRI/2022/09/045521 (status unknown) | 1. Vitamin E & C for 5 days (dose unspecified) 2. Standard of care | 84 | India | Children 5–18 y.o with dengue and PLT < 150 G/L *Excluded:* prior vitamin E/C use, need for platelet transfusion, severe GI bleeding. | NS1/IgM/ IgG | PLT count | Not available | NA |
| CTRI/2019/09/021244 (status unknown) | 1. Vitamin C* IV TID and B1 IV 100mg BID given as separate infusions over 30 min for 3 days 2. Standard of care *Dose: 1–3 yrs-130mg TID, 4–8yrs- 200mg TID, 9–13yrs- 400mg TID,14–18yrs- 600mg TID | 66 | India | Children with severe dengue (WHO 2009) *Excluded:* On steroids >2 weeks during admission, pre-existing liver/ renal/ haematological/cardiac disease. | Serology (unspecified) | Duration of IV fluid; duration of PICU stay; mortality | Not available | NA |
| SLCTR/2017/028 (status unknown) | 1. Vitamin C PO 1000mg BID for 5 days 2. Placebo | 60 | Sri Lanka | Patients 12–70 y.o with dengue within 3 days of fever onset, and PLT > 100G/L *Excluded*: plasma leakage/bleeding, severe vomiting/inability to take oral drugs, drowsiness, liver failure, AST/ ALT > 100 U/L, comorbidities, taking statins, beta blockers), severe dehydration, BMI > 27 kg/m², fatty liver disease, or use of alternative medicine. | NS1 | Hospital LOS; Proportion developing plasma leakage; Secondary bacterial infection | Not available | NA |
| *Miscellaneous* | | | | | | | | |
| SLCTR/2024/015 (recruiting) | 1. Link Natural Sudarshana (LNS) 1500mg PO TID for 10 days 2. LNS 1100mg PO TID for 10 days 3. Placebo | 240 | Sri Lanka | Adults 18–70 y.o with dengue within 48 hours of fever onset *Excluded*: Fluid leakage, pregnancy, allergy to LNS, known hepatic/ renal impairment, oral drug intolerance | NS1 PCR | Proportion of patients who progress to critical phase | Not available | NA |

*(Continued)*

**Table 6.** (Continued)

| Study | Intervention(s) & Comparison(s) | Sample size | Country | Patient population | Confirmatory test | Primary endpoint(s) | Results | Overall risk of bias (*) |
|---|---|---|---|---|---|---|---|---|
| CTRI/2021/06/034040 | 1. AQCH 200 mg PO TID for 7 days 2. AQCH 400 mg PO TID for 7 days 3. AQCH 600 mg PO TID for 7 days 4. Placebo | 676 | India | Adults 18–65 y.o with uncomplicated DF within 3 days of fever onset and PLT > 100G/L. *Excluded*: warning signs, DHF/DSS; pregnancy/lactation; co-existing febrile illness; significant underlying medical conditions; transaminases > 5x ULN; recent antiplatelet/anticoagulant drugs, recent blood transfusion | NS1 | Proportion of patients progressing to DWS OR severe dengue | Not available | NA |
| ITMCTR2100004502 | 1. Ganghuo Kangan granules 2. Ganghuo Kanggan granule simulator | 500 | China | Adults 18–65 y.o with dengue fever within 5 days of symptoms onset and no prior traditional Chinese medicine treatment. | | Time to complete fever reduction | Not available | NA |
| CTRI/2020/01/022785 | 1. HPLT031707 Syrup PO for 7 days or until PLT > 150G/L (dose by kg) | 44 | India | Children 2–12 y.o with dengue, PLT 50–100 G/L and ALT < 2xULN. *Excluded*: Grade 3/4 dengue, significant bleeding or PLT < 50G/L; diabetes, cardiovascular disease, haematological disorders, < 10 kg or >45 kg. | NS1/IgM | PLT count | Not available | NA |
| CTRI/2019/01/017096 | 1. Ganjhuvir Liquid (Desmodium gangeticum root extract) 10 ml BID (duration not specified) 2. Standard of care | 50 | India | Adults 18–60 y.o with grade 1 dengue, PLT 30–100 G/L and ALT < 165U/L. *Excluded:* DHF grade 3 or 4, or PLT < 30 G/L; pregnancy/lactation; haematological disorders; recent blood transfusion; significant renal impairment; HIV infection | Not specified | PLT count | Not available | NA |

*(Continued)*

| Study | Intervention(s) & Comparison(s) | Sample size | Country | Patient population | Confirma-tory test | Primary end-point(s) | Results | Overall risk of bias (*) |
|---|---|---|---|---|---|---|---|---|
| CTRI/2017/12/010834 | Antivirals*:<br>1. Chloroquine 5mg/kg PO 2mg/kg IM/IV<br>2. Doxycycline PO/IV 100mg BID<br>3. Prochlorpera-zine 5mg PO BID/12.5mg IV/IM BID<br>4. S-Adenosyl L-Methionine 400mg PO BID<br>5. Ribavirin 200mg PO QID<br>Anti-Cytokines*:<br>1. Chloroquine 5mg/kg PO; 2mg/kg IM/IV<br>2. Doxycycline 100mg PO/IV BID<br>3. Montelukast 10mg PO OD<br>4. Tolfenamic Acid 200mg PO BID<br>*various combina-tions of antivirals/anti-cytokines based on dengue fever grade. | 350 | India | Patients of all ages with dengue and more than 3 days of illness<br>*Excluded*: confirmed co-infections (malaria, Chikungunya, influenza). | NS1 and IgM and IgG and PCR | Recovery | Not available | NA |
| CTRI/2017/11/010586 | 1. AYUSH PJ-7: 700mg trial formulation<br>2. Tablets of 350mg BID for 10 days<br>3. Placebo | 250 | India | Patients with dengue<br>*Excluded:* severe CKD, CHF, advanced HIV, malignancy; pregnancy/lactation; anticoagulants, steroids, or immunosup-pressants; active intes-tinal disorders/severe allergies | NS1/IgM | Effect on capillary leak | Not available | NA |
| CTRI/2017/11/010547 | 1. J P Nutrition Per Rectal: 2 ml per kg twice a day (<15 years); 60 ml (adults) | 100 | India | Dengue | NS1 | Symptom-atic relief | Not available | NA |
| ChiCTR-IPR-16009233 | 1. Lianbizhi injection plus standard of care<br>2. Standard of care<br>*SOC includes ibuprofen, diaze-pam, Vitamin C and Vitamin B6 | 120 | China | Patients 14–75 y.o with dengue<br>*Excluded:* Severe den-gue; uncontrolled comor-bidities: diabetes, heart disease, liver cirrhosis; renal insufficiency; secondary infection; pregnancy; known drug allergies | NS1/IgM | Fever relief time | Not available | NA |

*(Continued)*

**Table 6.** (Continued)

| Study | Intervention(s) & Comparison(s) | Sample size | Country | Patient population | Confirmatory test | Primary endpoint(s) | Results | Overall risk of bias (*) |
|---|---|---|---|---|---|---|---|---|
| ChiCTR-IPR-15006778 | 1. Tanreqing injection<br>2. Standard of care | 316 | China | Adults 18–65 y.o with dengue and fever >38°C for less than 48 hours. *Excluded:* Severe bleeding; DSS; severe liver/renal injury, ARDS; pregnancy/ lactation; allergies | NS1/PCR | Duration of fever | Not available | NA |
| ChiCTR-TRC-14005244 | 1. Test group: ReDuNing injection<br>2. Control group: Western medical treatments | 72 | China | Patients 16–70 y.o with probable dengue (fever >38.5°C, clinical symptoms, and epidemiological risk/cytopenias) *Excluded:* Severe dengue; secondary infection; diabetes, heart/liver/renal disease, COPD, G6PD deficiency; obesity/ severe malnutrition; allergies | None required | Time to fever resolution | Not available | NA |

(*) Studies assessed as having a overall low risk of bias are shaded as green, some risk of bias as yellow and high or critical risk of bias as red.

Abbreviations: ALT alanine aminotransferases; AQCH tabs contents undefined; AST aspartate aminotransferase; AYUSH tabs contents undefined; BID twice daily; BMI body mass index; CHF congestive heart failure; CKD chronic kidney disease; COPD chronic obstructive pulmonary disease; DF dengue fever; DHF dengue haemorrhagic fever; DSS dengue shock syndrome; DWS dengue with warning signs; G6PD glucose-6-phosphate dehydrogenase deficiency; HIV human immunodeficiency virus; IgG immunoglobulin G; IgM immunoglobulin M; ITP Immune thrombocytopenia; IV intravenously; LOS length of stay; NS1 non-structural protein 1; NSAID nonsteroidal anti-inflammatory drug; OD once daily; PCR polymerase chain reaction; PLT platelet count; PO by mouth; T temperature; TID three times daily; ULN upper limit of normal; WBC white blood cells; WHO World Health Organisation.

(Fig 4). These included duration of detectable viraemia (2/8, 25%), change in absolute viral load (4/8, 50%), and log-transformed viral load area under the curve (AUC, 2/8, 25%). Measurement methods and time points varied between studies: one assessed infectious virus titres using the NSET assay (CTRI/2021/07/035290), while the others quantified viral RNA by RT-PCR. Two studies used time to NS1 antigen clearance as the primary endpoint, defined as the interval from treatment initiation to the first of two consecutive NS1 ELISA-negative plasma samples.

Five trials (50%) used clinical primary endpoints, although definitions differed. Two studies assessed efficacy based on fever: one measured time to fever clearance, while the other used area under the curve (AUC) for temperature above 37°C. Safety endpoints also varied, with most reporting adverse event frequency. Only one study evaluated the proportion of participants developing DHF, defined as platelet count below 100,000/mm$^3$ and evidence of plasma leakage (≥20% rise in haematocrit from baseline or presence of pleural effusion).

**Primary endpoints used in trials of host-directed therapies.** The choice of primary endpoint in trials investigating HDT (Tables 3 and 4, n = 34) was highly heterogeneous (Fig 4). Overall, platelet count – measured using various metrics - was the most frequently reported endpoint (15/34, 44%), followed by mortality (7/34, 21%), plasma leakage (5/34, 15%), adverse events (5/34, 15%), DSS (4/34, 12%) and inflammatory markers (4/34, 12%). Less commonly reported endpoints (1/34 each, 3%) included dengue with warning signs, duration of illness and hospital stay, modified Sequential Organ Failure Assessment and ICU admission. Trials enrolling adults or mixed populations (n = 27) most often used platelet-related endpoints, whereas paediatric trials (n = 7) prioritised critical outcomes like mortality and dengue shock syndrome (DSS) (Fig 5). None of the trials enrolling patients with severe dengue (n = 8) used platelet measurements as primary endpoints.

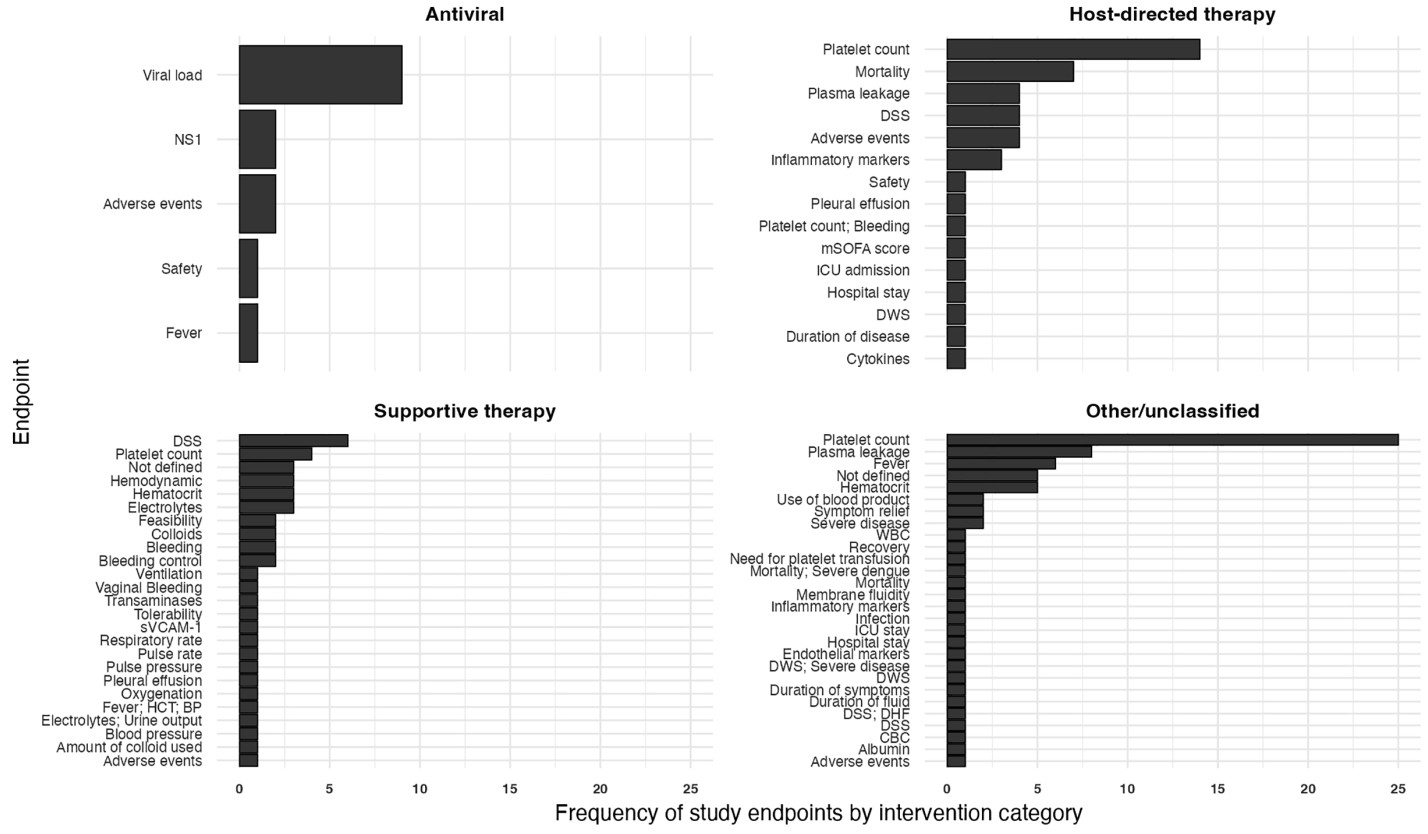

**Fig 4. Frequency of primary endpoints across dengue interventional trials, categorised by proposed mechanism of action of intervention.**

Approaches to measuring outcomes also varied substantially (Fig 5). Plasma leakage was assessed using different combinations of criteria, including the presence of pleural effusion or ascites on imaging—measured by MRI in one study and ultrasound in others—alongside increases in haematocrit or plasma biomarkers (e.g., syndecan-1). Even for laboratory-based endpoints such as platelet count, diverse summary measures were applied, including absolute or relative change, nadir value, and the proportion of participants above or below prespecified thresholds.

**Primary endpoints used in trials evaluating _Carica papaya_ leaf extract, micronutrients and 'miscellaneous' treatments.** The other studies (Tables 5 and 6, n = 46) also employed a wide range of primary endpoints (Fig 4). Platelet count was again the most frequently used primary endpoint (25/46, 51%). Plasma leakage, defined by various criteria on ultrasound—with or without accompanying increases in haematocrit—was assessed in 6 studies (13%). Changes in haematocrit were specifically designated as the primary outcome in 5 studies (10.8%). Time to fever resolution was used in 6 trials (13%), although definitions of this endpoint were often unclear. The remaining endpoints were used in only 1 or 2 studies and included duration of illness, serum chloride and bicarbonate levels, clotting assays, endothelial biomarkers, requirement for intravenous fluids, liver enzyme levels, respiratory rate and treatment tolerability.

## Discussion

Despite the considerable morbidity and public health burden caused by dengue, this review highlights the remarkably barren treatment landscape for patients with symptomatic infection. We identified very few clinical trials evaluating

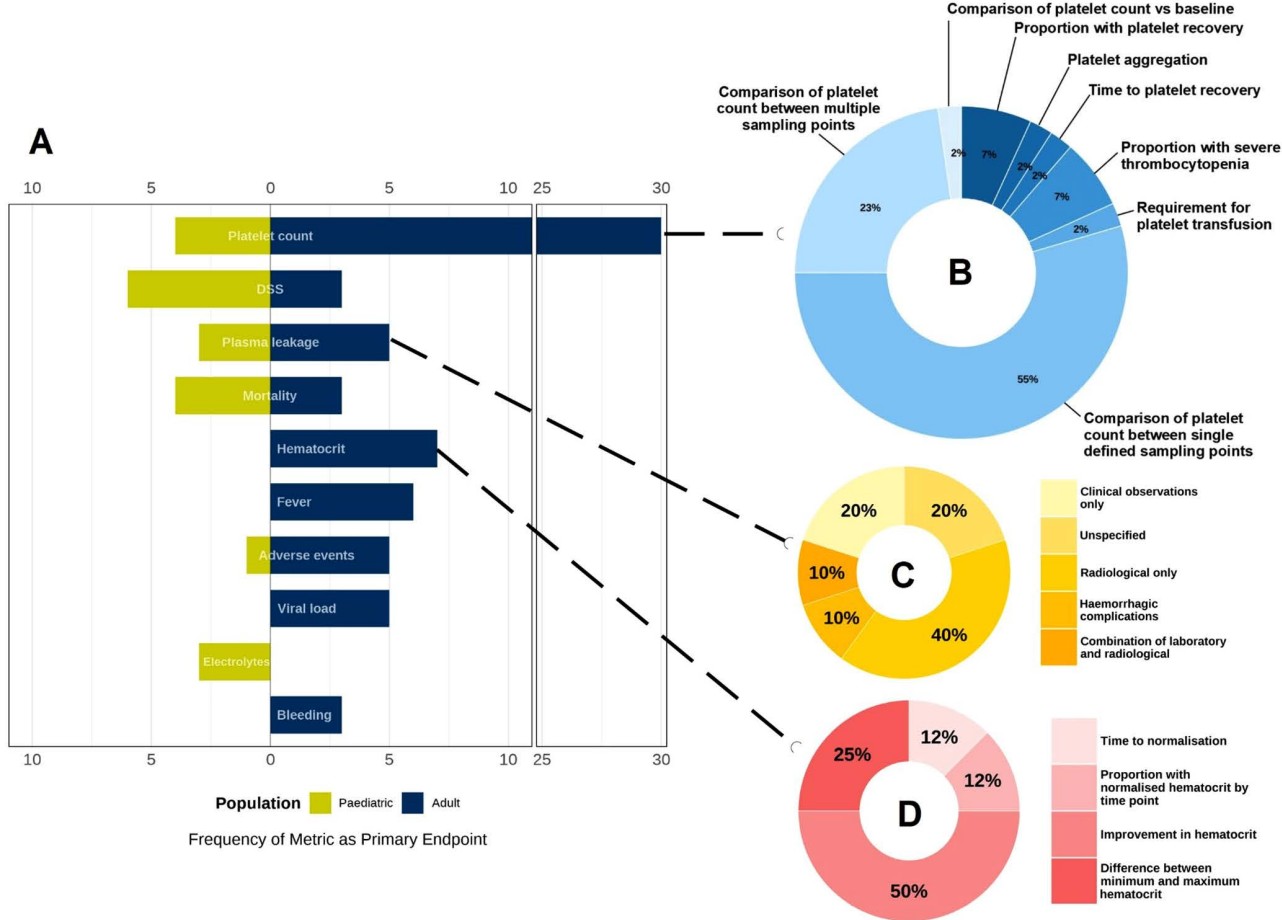

**Fig 5. Visualisation of heterogeneity in primary endpoints used in dengue interventional trials.**

therapeutics with proposed antiviral or host-directed mechanisms of action, and no single intervention had a sufficiently sized, consistent or high-quality evidence base to make recommendations for use in clinical practice. Most published studies were small, underpowered to detect clinically meaningful outcomes, and many were assessed as having a high or critical risk of bias.

## Antivirals

Our review identified only five published randomised trials evaluating therapeutic agents with putative antiviral activity against dengue. All five trials evaluated repurposed therapeutics originally developed for other indications, none recruited participants under 15 years of age, and none demonstrated significant antiviral activity.

Whether antiviral agents can be delivered early enough, and at concentrations sufficient to meaningfully reduce viral load within the narrow window between symptom onset and immune-mediated clearance, remains uncertain. Progress in this field will likely depend on the development of dengue-specific direct-acting antivirals, rather than relying on repurposed treatments. Trials of potentially potent dengue antiviral agents have recently suffered from early termination due to industry deprioritisation of dengue research. However, there are signs of renewed momentum, with novel drug discovery efforts [71–73] and ongoing phase 2 trials including a dengue-specific monoclonal antibody (SII, CTRI/2021/07/035290),

a trial of molnupiravir for dengue and chikungunya in Brazil (Brazilian Clinical Trials Registry: U1111-1306–1425) and a small-molecule NS4B inhibitor (Novartis-EYU688). Sustaining this momentum will be essential to reinvigorate the neglected antiviral dengue therapeutics pipeline. Inclusion of paediatric populations in future trials is also critical for health equity, given that children are still bearing a substantial burden in hyper-endemic settings such as Vietnam [74].

## Corticosteroids

Although corticosteroids were the most frequently evaluated treatment for dengue, we identified only 944 patients across nine published studies, encompassing a wide spectrum of clinical phenotypes from early symptomatic dengue to refractory shock. Among patients with dengue shock, no trial demonstrated a mortality benefit with corticosteroid treatment; however, all were underpowered to evaluate this endpoint. While we did not perform a meta-analysis, a 2014 Cochrane review similarly found no effect of corticosteroids on mortality among 284 paediatric patients with dengue shock. However, based on an assumed 21.3% risk of death due to dengue shock, the authors estimated that a definitive trial would require more than 1600 participants to detect a 25% mortality reduction [15]. Recent trends in dengue shock mortality have been mixed: rates have declined in some settings with improved supportive care and monitoring but increased in others [3] particularly among patients with obesity, comorbidities, or limited access to expert clinical management. These observations underscore the importance of a large, adequately powered, multi-site trial with careful patient selection to evaluate corticosteroid efficacy in severe dengue.

In patients with non-severe dengue, corticosteroid therapy did not affect clinical or laboratory outcomes, but the studies were also underpowered. We consider that routine use of corticosteroids in early uncomplicated dengue would be hard to justify, given the low risk of progression to severe disease among unselected outpatients.

Despite the paucity of evidence, corticosteroids continue to be used empirically in many endemic settings, particularly for critically ill patients [75–77]. Experience from COVID-19 demonstrates that corticosteroids can reduce mortality in specific subgroups of patients with severe viral infections, but harm can also outweigh benefits in other groups [78]. Robust, adequately powered clinical trials are therefore essential to determine whether corticosteroids confer benefit—or potential harm—in moderate and severe dengue, and to explore whether treatment effects vary according to baseline characteristics or disease severity.

## Other therapeutics

Given the variable formulations and methodological limitations of published studies, the current evidence base is insufficient to draw conclusions regarding the clinical efficacy of CPLE in acute dengue. Further low-quality studies using non-standardised preparations and surrogate laboratory endpoints are unlikely to add meaningfully to the evidence base. We recognise, however, the considerable regional interest and widespread informal use of CPLE across South and Southeast Asia and Indonesia. Formal evaluation of the proposed active compound is therefore warranted, but should begin with in vitro studies, followed by rigorously designed human trials using pharmaceutical-grade material in randomised, placebo-controlled settings with clinically relevant endpoints.

Among the other host-directed therapeutics, few trials were assessed as being at low risk of bias. Well-designed trials investigating agents such as rupatadine and montelukast have shown no significant clinical benefit despite using appropriate endpoints [33,34,36]. Trials with a higher risk of bias frequently lacked clear therapeutic rationale and focused primarily on reporting on laboratory outcomes such as platelet counts. In several cases, investigators proposed different mechanisms of action for the same agent – for example, doxycycline was hypothesised to act through both antiviral and host-directed pathways in separate studies [20,43]. These limited successes highlight persistent challenges in dengue clinical research: incomplete understanding of disease pathogenesis, difficulties in patient selection and the absence of standardised endpoints. While recent work has linked hyperinflammatory responses with adverse outcomes, no single, universally accepted pathway has been identified as the key driver of severe disease. This uncertainty complicates

rational target selection. Moreover, the clinical heterogeneity of dengue, coupled with the lack of reliable predictors of disease progression, often results in trials enrolling mixed patient populations, potentially diluting treatment effects.

Overcoming these challenges will require innovation in both trial design and patient stratification. Adaptive and platform trial designs could enable concurrent evaluation of multiple candidate therapies—an efficient approach in the absence of established treatments. Future studies should incorporate biomarker-based or multi-omics approaches to identify subgroups most likely to benefit from targeted interventions. Although most existing trials have been underpowered to demonstrate clinical efficacy, mechanistic studies of anti-inflammatory or endothelial-stabilising therapies, when conducted with methodological rigour and harmonised endpoints, can still provide valuable insights and support future meta-analyses.

### Endpoints

This review underscores the marked heterogeneity of primary endpoints used in dengue therapeutic trials. To date, no large clinical trial of either antiviral or host-directed therapy has been adequately powered to detect clinically meaningful effects on key outcomes such as hospitalisation (for early treatment) or mortality (for late treatment). Because only a small proportion of patients progress to severe disease, many early-phase studies have relied on surrogate endpoints to assess efficacy. However, there are currently no universally accepted surrogate outcomes for dengue, limiting comparability across trials.

Selecting appropriate endpoints poses particular challenges for antiviral trials. Viraemia peaks within the first few days of illness, whereas most patients are hospitalised later during the critical phase, when viral replication is already declining. This narrow therapeutic window is further complicated by variability in viraemia kinetics and viral clearance, which depend on immune status and infecting serotype [9]. Fever duration has been used as a pragmatic endpoint; however, it functions merely as a clinical proxy for viremia, and accurate determination requires frequent measurement, and early presentation, which is uncommon for a disease generally considered self-limiting.

Trials of other therapeutics have often used platelet count–based metrics as primary endpoints. However, these measures are poor surrogates for clinical benefit, reflecting health-seeking behaviour and disease pathophysiology rather than treatment response. Patients typically present during the critical phase—at or near the platelet nadir—leaving little opportunity for an intervention to alter trajectory [79]. Consequently, absolute platelet counts are often dissociated from clinical outcomes.

More robust clinical outcomes, such as progression to severe dengue or mortality are difficult to use because they require very large sample sizes. Intermediate endpoints, such as warning signs or plasma leakage, are also limited by subjectivity and lack of standardisation; for example, fluid accumulation assessments depend heavily on operator technique and imaging modality. Although earlier efforts sought to standardise definitions for clinical endpoints [80], they have not achieved widespread adoption. As a result, the absence of uniformly defined clinical and virological outcomes has hampered meta-analyses and synthesis of existing data.

To address this gap, a standardised core outcome set has recently been developed through the DEN-CORE initiative [14]. This global consensus, involving stakeholders from 36 countries including patient representatives, defined a Core Outcome Set (COS) and Core Outcome Measurement Set (COMS) for dengue therapeutic trials. The group prioritised seven core outcomes for hospitalised patients and eleven for those with early symptomatic dengue. Adoption of this core outcome framework in all future dengue trials will address the long-standing inconsistencies in outcome selection and reporting, enabling meaningful data comparison and evidence synthesis across studies.

In line with global recognition of the growing dengue burden, the WHO has recently published Target Product Profiles (TPPs) for dengue therapeutics, defining the minimum and optimal characteristics required for treatments to be suitable for public health use [81]. Future drug discovery efforts and clinical trials should align with the TPPs to ensure that treatment candidates are developed with standardized, globally agreed benchmarks for both adults and children.

Table 7 summarises the key gaps identified from this systematic review and implications for future research.

**Table 7. Key gaps identified from this systematic review and implications for future research.**

| Key gap identified | Evidence from this review | Implications for future research |
|---|---|---|
| **Study design and methods** | | |
| Small sample sizes limit detection of clinically meaningful outcomes | Most studies were small and underpowered | Platform or collaborative multi-stage multi-centre trial designs: <br> -To allow quicker "scan" of candidates in phase 2 trials <br> -To move safe and potential candidates into large phase 3 trials for clinically meaningful outcomes |
| Very few adequately powered phase 3 trials | Only 11/72 (15% published studies) enrolled ≥250 participants | |
| High or critical risk of bias common | 24/52 (45%) published trials had high/critical risk of bias due to randomisation, allocation concealment, blinding methods or lack of those information | -Mandate centralized, computer-generated randomization <br> -Use of central pharmacy-controlled or sponsor-held allocation <br> -Compliance with standardized frameworks for clinical trials, including prospective trial registration and international reporting guidelines <br> -Partner with regional trial networks for concealment support where infrastructure is limited |
| **Endpoint heterogeneity** | | |
| High variability in endpoint selection | Highly heterogeneous primary endpoints across antiviral and host directed therapeutic trials | Adoption of a consensus core outcome set (e.g., DEN-CORE) is critical |
| High variability in endpoint measurement methods | Plasma leakage assessed using different modalities and criteria | Standardised definitions and measurement protocols are needed |
| Frequent use of non - clinical outcomes | Platelet count was the most common primary endpoint | Platform or collaborative multi-centre trial design would enable adequate power to evaluate clinically meaningful outcomes, including progression to severe dengue, hospitalisation, mortality. |
| **Limited meta-analysis feasibility** | | |
| Inability to pool data | Non-standardised endpoints and risk of bias concerns precluded meta-analysis | Harmonised design and data standards are required to enable individual level meta-analysis |
| **Fragmented therapeutic pipelines** | | |
| Many single-agent, single-study evaluations | 53 compounds evaluated in only 1–3 small studies each | Coordinated prioritisation (e.g., in alignment with WHO TPPs) and adaptive platform trials could reduce duplication and inefficiency |
| Absence of paediatric participants in antiviral trials | No antiviral trials included participants <15 years | Future antiviral studies must include children (in line with WHO TPPs) to ensure equity and relevance to endemic settings |

## Strengths and limitations

To our knowledge, this is the first systematic mapping review to comprehensively chart the landscape of dengue treatment trials. We conducted an extensive search in electronic databases and clinical trial registries, including both non-randomized studies alongside randomized controlled trials to ensure a comprehensive view of the evidence base. We also flagged ongoing studies to evaluate potential future shifts in therapeutic approaches.

However, several limitations exist. First, we only review English-language articles due to limited translation resources. This restriction might exclude relevant local data published in national journals from Latin American or other non-English speaking countries. Secondly, regarding the review process, while title, abstract, and full-text screening were performed independently by two reviewers to minimize selection error, data extraction was mainly performed by a single reviewer due to resource constraints. To mitigate potential inaccuracies, a random sample of the included studies underwent double data entry to assess the degree of agreement and identify systematic discrepancies before proceeding with single entry for the remaining studies. We also had a second reviewer verify all extracted primary endpoints and perform a comprehensive data cleaning process prior to analysis. Furthermore, we believe this approach does not materially affect the study's conclusions, as meta-analysis was precluded due to the heterogeneity of study characteristics. Finally, regarding quality assessment, it is noted that we utilized the version of ROBINS-I available prior to the October 2025 update.

## Conclusion

Our findings demonstrate the striking neglect of dengue in the field of therapeutic research, despite its substantial global burden and disproportionate impact on low- and middle-income countries. There is an urgent need for large, adequately powered clinical trials that use harmonised and clinically meaningful endpoints to generate robust evidence for treatment and inform policy. Innovative trial designs – such as adaptive platform trials employing factorial randomisation – offer an efficient and collaborative strategy to evaluate multiple candidate therapies concurrently. Prioritising equitable participation, including representation from a range of dengue-endemic regions and vulnerable populations including children, will be essential to ensure that future discoveries translate into accessible, evidence-based care for all patients affected by dengue.

## Supporting information

**S1 Fig. Numbers of adults (≥ 16 years old) and children (<16 years old) enrolled in each therapeutic category.**
(TIFF)

**S1 PRISMA Checklist. Preferred Reporting Items for Systematic Reviews and Meta-Analyses (PRISMA)**. From: Page MJ, McKenzie JE, Bossuyt PM, Boutron I, Hoffmann TC, Mulrow CD, et al. The PRISMA 2020 statement: an updated guideline for reporting systematic reviews. BMJ 2021;372:n71. https://doi.org/10.1136/bmj.n71.
(PDF)

**S1 File. Search terms used in the systematic review.**
(PDF)

**S1 Table. Risk of bias assessments of the included publications.**
(PDF)

**S1 Data. Detailed risk of bias assessments.**
(ZIP)

**S2 Data. Data extraction and logging.**
(ZIP)

**S3 Data. Studies screened included excluded.**
(ZIP)

## Acknowledgments

We would like to thank Ms Kalynn Kennon, Head of Data Engineering, Infectious Diseases Data Observatory for her support with this project.

## Author contributions

**Conceptualization:** Tran Bang Huyen, Caitlin Naylor, Nguyen Lam Vuong, Po-Ying Chia, Phung Khanh Lam, Sophie Yacoub.

**Data curation:** Tran Bang Huyen, Angela McBride, Tun-Linn Thein, Khoi Minh Le, Elinor Harriss, Matthew JW Kain, Nguyen Lam Vuong, Po-Ying Chia, Phung Khanh Lam, James Andrew Watson, Sophie Yacoub.

**Formal analysis:** Tran Bang Huyen, Angela McBride, Khoi Minh Le.

**Funding acquisition:** Tran Bang Huyen, Phung Khanh Lam, Sophie Yacoub.

**Investigation:** Tran Bang Huyen, Angela McBride, Tun-Linn Thein, Khoi Minh Le, Tran Luu, Nguyen Quang Huy, Matthew JW Kain, Jonathan Cattrall, Ho Quang Chanh, Nguyen Lam Vuong, Phung Khanh Lam, Sophie Yacoub.

**Methodology:** Tran Bang Huyen, Angela McBride, Tun-Linn Thein, Caitlin Naylor, Daniel Munblit, Po-Ying Chia, Phung Khanh Lam, James Andrew Watson, Sophie Yacoub.

**Project administration:** Tran Bang Huyen, Angela McBride.

**Software:** Caitlin Naylor.

**Supervision:** Phung Khanh Lam, James Andrew Watson, Sophie Yacoub.

**Validation:** Angela McBride.

**Visualization:** Tran Bang Huyen, Khoi Minh Le, Matthew JW Kain.

**Writing – original draft:** Tran Bang Huyen, Angela McBride.

**Writing – review & editing:** Tran Bang Huyen, Angela McBride, Tun-Linn Thein, Khoi Minh Le, Tran Luu, Nguyen Quang Huy, Matthew JW Kain, Jonathan Cattrall, Caitlin Naylor, Ho Quang Chanh, Nguyen Lam Vuong, Daniel Munblit, Po-Ying Chia, Phung Khanh Lam, James Andrew Watson, Sophie Yacoub.

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
