## [Decision Letter · Decision Letter 0]

22 Feb 2026

PNTD-D-25-02236

A systematic mapping review of therapeutic clinical trials in dengue

Dear Dr. Huyen,

Thank you for submitting your manuscript to PLOS Neglected Tropical Diseases. After careful consideration, we feel that it has merit but does not fully meet PLOS Neglected Tropical Diseases's publication criteria as it currently stands. Therefore, we invite you to submit a revised version of the manuscript that addresses the points raised during the review process.

We look forward to receiving your revised manuscript.

Kind regards,

Kentaro Iwata

Academic Editor

Michael Holbrook

Section Editor

Shaden Kamhawi

co-Editor-in-Chief

Paul Brindley

co-Editor-in-Chief

**Additional Editor Comments:**

Please read and respond to the reviewer's comments and suggestions, and resubmit an improved version.

**Journal Requirements:**

At this stage, the following Authors/Authors require contributions: Tran Bang Huyen. Please ensure that the full contributions of each author are acknowledged in the "Add/Edit/Remove Authors" section of our submission form.

3) We have noticed that you have a list of Supporting Information legends in your manuscript. However, there are no corresponding files uploaded to the submission. Please upload them as separate files with the item type 'Supporting Information'.

4) When completing the data availability statement of the submission form, you indicated that you will make your data available on acceptance. We strongly recommend all authors decide on a data sharing plan before acceptance, as the process can be lengthy and hold up publication timelines. Please note that, though access restrictions are acceptable now, your entire data will need to be made freely accessible if your manuscript is accepted for publication. This policy applies to all data except where public deposition would breach compliance with the protocol approved by your research ethics board. If you are unable to adhere to our open data policy, please kindly revise your statement to explain your reasoning and we will seek the editor's input on an exemption. Please be assured that, once you have provided your new statement, the assessment of your exemption will not hold up the peer review process.

5) As required by our policy on Data Availability, please ensure your manuscript or supplementary information includes the following:

6) Please ensure that the funders and grant numbers match between the Financial Disclosure field and the Funding Information tab in your submission form. Note that the funders must be provided in the same order in both places as well.

**Reviewers' comments:**

Reviewer's Responses to Questions

**Key Review Criteria Required for Acceptance?**

**Methods**

-Are the objectives of the study clearly articulated with a clear testable hypothesis stated?

-Is the study design appropriate to address the stated objectives?

-Is the population clearly described and appropriate for the hypothesis being tested?

-Is the sample size sufficient to ensure adequate power to address the hypothesis being tested?

-Were correct statistical analysis used to support conclusions?

-Are there concerns about ethical or regulatory requirements being met?

Reviewer #1: See below

**Results**

-Does the analysis presented match the analysis plan?

-Are the results clearly and completely presented?

-Are the figures (Tables, Images) of sufficient quality for clarity?

Reviewer #1: See below

**Conclusions**

-Are the conclusions supported by the data presented?

-Are the limitations of analysis clearly described?

-Do the authors discuss how these data can be helpful to advance our understanding of the topic under study?

-Is public health relevance addressed?

Reviewer #1: See below

**Editorial and Data Presentation Modifications?**

Reviewer #1: See below

**Summary and General Comments**

Reviewer #1: The manuscript presents a relevant and timely systematic review mapping the landscape of clinical trials of dengue therapies published and registered. The findings effectively support the central message: the evidence is fragmented, with low standardization of outcomes and often limited methodological quality, which hinders meta-analyses and delays the advancement of therapies. In terms of contribution, the work has merit and may help guide future studies in dengue treatment.

Some key strengths are the relevant and well-justified question, the road coverage of sources (MEDLINE/EMBASE + registries -ICTRP and ClinicalTrials.gov- with updated search), duplicate screening with a standardized tool (Covidence) and appropriate emphasis on outcomes and quality.

I have minor comments and some concerns:

1. According to the abstract, the 31 studies related to supportive treatment were excluded. However, they were included and reviewed for primary endpoint, as stated in L174-176, and later excluded (for quality synthesis and design). Please clarify and inform if this was adjusted pos hoc.

2. L179 - Text includes a highlighted sentence of error instead of citing the Figure (2A?)

3. Fig 2A: What “Latin American” means? Other countries besides the ones already nominated?

4. Line 412- The authors afirm that children represent “major”risk group. It may vary according to regions. Please comment and include references.

5. L450-451- provide references

6. L484-485: “Fever duration ... is subjective ...”. It can be objectively measured and even daily monitored using eletronic devices.

7. It would be interesting to have a “Top gaps” table. Ex: o few phase 3 studies, absence of pediatrics in antivirals, non-clinical outcomes, heterogeneity of plasma leakage measurements, recurring randomization/blinding/loss problems.

8. It would strengthen the Discussion to assess how frequently tested candidate therapies map onto the WHO dengue treatment TPPs.

9. The manuscript states that articles were extracted at two time points (2023 and 2024). However, the data dictionary in the appendix includes notes indicating that some endpoints/fields were added only in 2024. Could the authors clarify whether the full set of fields was applied retrospectively to all articles extracted in 2023 (i.e., were earlier extractions updated)? If not, is there a risk that some variables were not collected for a subset of articles, potentially leading to missingness or inconsistent endpoint availability across the dataset?

PLOS authors have the option to publish the peer review history of their article (what does this mean?). If published, this will include your full peer review and any attached files.

**Do you want your identity to be public for this peer review?** For information about this choice, including consent withdrawal, please see our Privacy Policy.

Reviewer #1: **Yes:**Viviane Boaventura

**Figure resubmission:**
---

## [Decision Letter · Decision Letter 1]

14 May 2026

Dear Huyen,

We are pleased to inform you that your manuscript 'A systematic mapping review of therapeutic clinical trials in dengue' has been provisionally accepted for publication in PLOS Neglected Tropical Diseases.

Best regards,

Kentaro Iwata

Academic Editor

Michael Holbrook

Section Editor

Shaden Kamhawi

co-Editor-in-Chief

Paul Brindley

co-Editor-in-Chief

Reviewer's Responses to Questions

**Key Review Criteria Required for Acceptance?**

**Methods**

-Are the objectives of the study clearly articulated with a clear testable hypothesis stated?

-Is the study design appropriate to address the stated objectives?

-Is the population clearly described and appropriate for the hypothesis being tested?

-Is the sample size sufficient to ensure adequate power to address the hypothesis being tested?

-Were correct statistical analysis used to support conclusions?

-Are there concerns about ethical or regulatory requirements being met?

Reviewer #1: Yes

Reviewer #2: yes

**Results**

-Does the analysis presented match the analysis plan?

-Are the results clearly and completely presented?

-Are the figures (Tables, Images) of sufficient quality for clarity?

Reviewer #1: Yes

Reviewer #2: yes

**Conclusions**

-Are the conclusions supported by the data presented?

-Are the limitations of analysis clearly described?

-Do the authors discuss how these data can be helpful to advance our understanding of the topic under study?

-Is public health relevance addressed?

Reviewer #1: Yes

Reviewer #2: yes

**Editorial and Data Presentation Modifications?**

Reviewer #1: All questions were properly answered. I suggest accepting the manuscript.

Reviewer #2: Authors improved their paper and now can be accept

**Summary and General Comments**

Reviewer #1: The revisions effectively addressed all the issues.

Reviewer #2: Authors improved their paper and now can be accept

PLOS authors have the option to publish the peer review history of their article (what does this mean?). If published, this will include your full peer review and any attached files.

Reviewer #1: **Yes:**Viviane Sampaio Boaventura

Reviewer #2: **Yes:**Francesco Di Gennaro

---

## [Editor Report · Acceptance letter]

Dear Dr Huyen,

We are delighted to inform you that your manuscript, "A systematic mapping review of therapeutic clinical trials in dengue," has been formally accepted for publication in PLOS Neglected Tropical Diseases.

Best regards,

Shaden Kamhawi

co-Editor-in-Chief

Paul Brindley

co-Editor-in-Chief
